# AN X-RAY IS WORTH 15 FEATURES: SPARSE AUTOENCODERS FOR INTERPRETABLE RADIOLOGY REPORT GENERATION

## ABSTRACT

Radiological services are experiencing unprecedented demand, leading to increased interest in automating radiology report generation. Existing Vision-Language Models (VLMs) suffer from hallucinations, lack interpretability, and require expensive fine-tuning. Sparse Autoencoders (SAEs) have been shown to provide a principled approach to reverse-engineer a model's internal activations into discrete, verifiable components. Thus, we introduce SAE-Rad, the first instance of using mechanistic interpretability techniques explicitly for a downstream multi-modal reasoning task. SAE-Rad uses a novel SAE architecture to decompose latent representations from a pre-trained vision transformer into human-interpretable features. These features are then labelled using an off-the-shelf language model and compiled into a full report for each image, eliminating the need for fine-tuning large models for this task. On the MIMIC-CXR dataset, SAE-Rad achieves competitive radiology-specific metrics compared to state-of-the-art models while using significantly fewer computational resources for training. Qualitative analysis reveals that SAE-Rad learns meaningful visual concepts and generates reports aligning closely with expert interpretations. Our results suggest that SAEs can enhance multimodal reasoning in healthcare, providing a more interpretable alternative to existing VLMs.

## 1 INTRODUCTION

Radiological services are essential to modern clinical practice, with demand rising rapidly. In the UK, the NHS performs over 43 million radiological procedures annually (Lewis et al., 2021), costing over £2 billion, and demand for scans more than doubled between 2012 and 2019 (NHS England & NHS Improvement, 2019). Much of the cost covers agency, bank, and overtime staff, but a national imaging strategy deemed it unsustainable. (NHS England & NHS Improvement, 2019). Consequently, there's growing interest in (semi-)automating tasks like radiology report generation, augmentation, and summarization to assist clinicians (Zhu et al., 2024; Chen et al., 2024; Pérez-García et al., 2024), spurred by advances in multimodal text-vision modelling techniques.

Recent architectures combining vision encoders with pretrained Large Language Models (LLMs) into multimodal Vision-Language Models (VLMs) excel in visual and language tasks (Liu et al., 2024b; 2023a; Li et al., 2024; Lin et al., 2023; Liu et al., 2023b). VLMs have been applied to healthcare tasks, including radiology report generation (Hyland et al., 2023; Bannur et al., 2024; Chen et al., 2024; Stock et al., 2024; Yang et al., 2024), often by mapping image representations into the LLM's token embedding space (Pérez-García et al., 2024). Despite improvements from scaling VLMs (Yang et al., 2024), hallucinations and disagreements with domain experts remain common (Yildirim et al., 2024; Jeblick et al., 2023; Lee et al., 2023; Tanno et al., 2024a). Hallucinations are unavoidable in LLMs (Kalai & Vempala, 2024; Xu et al., 2024), and this limits VLMs for radiology reporting, but other important considerations remain. For current state-of-the-art systems, it is necessary to finetune a multi-billion parameter LLM to perform visual instruction tuning (Liu et al., 2024b), which is computationally intensive. Additionally, reports generated by VLMs may not be faithful to the underlying computations of the image encoder – we seek to design a framework that is verifiably faithful to the image model by reverse engineering it's computations. This may yield more interpretable results, engendering more trust in automated radiology reporting models.

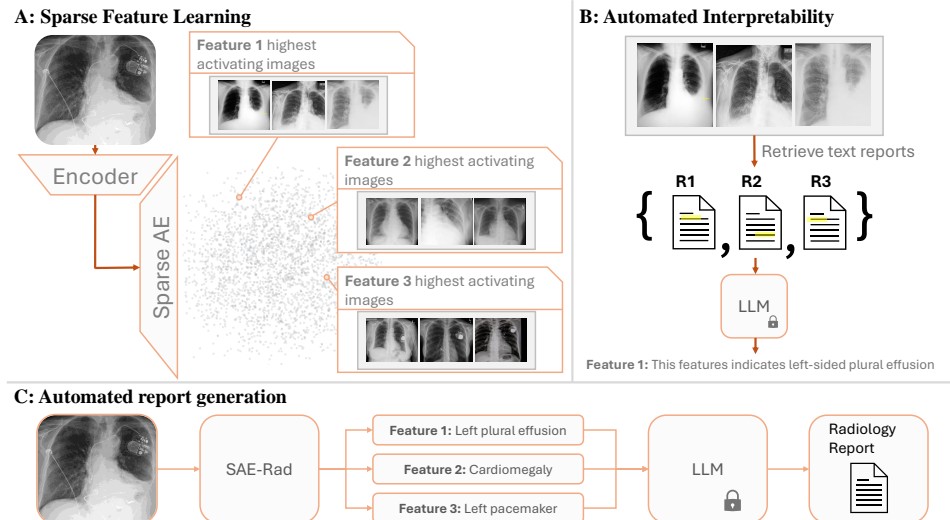

Figure 1: **SAE-Rad overview**. **Panel A:** We learn a set of sparsely activating features by training a Sparse Autoencoder (SAE) on class tokens produced by a radiology-image encoder. **Panel B:** We retrieve the corresponding reference reports for highest activating images for a feature, from which we can produce text descriptions of each feature. **Panel C:** We pass a new image through the radiology-image encoder and SAE encoder to retrieve the highest activating features. Text descriptions of these features are subsequently used by a pretrained large language model (LLM) to generate a detailed radiology report.

To this end, we introduce SAE-Rad (Fig. 1), a framework which leverages sparse autoencoders (SAEs) to learn human-interpretable features from image latents extracted from a pre-trained radiology image encoder (Pérez-García et al., 2024). We propose to automatically label these features using pre-trained LLMs (Bricken et al., 2023) and then compile them into descriptive radiology reports. To the best of our knowledge, this is the first instance of using SAEs explicitly for a downstream task requiring multi-modal reasoning. Our contributions are as follows: 1) *Novel SAE:* We propose a hybrid architecture which builds on gated SAEs to achieve accurate reconstructions with comparable sparsity to state-of-the-art techniques; 2) *Multimodal reasoning pipeline:* We develop a framework based on our SAE which leverages automated interpretability to label SAE features using paired text data and ultimately produce radiology reports given an image; 3) *Interpretability in healthcare:* Our case study on healthcare data demonstrates the utility of mechanistic interpretability for downstream tasks in a relatively smaller and significantly more homogeneous dataset than most general-domain text or natural-image datasets.

## 2 RELATED WORK

**Multimodal reasoning**    Multimodal reasoning methods like ScienceQA (Lu et al., 2022) introduced multimodal chain-of-thought (CoT) by zero-shot prompting models to generate both rationales and answers. Multimodal-CoT (MM-CoT) (Zhang et al., 2023) extended this with a two-stage framework that separates rationale generation and answer inference using two models of the same architecture. Duty-Distinct CoT (DDCoT) (Zheng et al., 2023) further factorizes rationales by decomposing the initial question into sub-questions answered by a vision-language model (VLM). Other divide-and-conquer approaches decompose questions into sub-questions but often require training task-specific visual question generation (VQG) models and additional scoring models (Selvaraju et al., 2020; Uehara et al., 2022; Wang et al., 2022). IdealGPT (You et al., 2023) iteratively decomposes queries and uses a VLM to answer sub-questions, repeating the process if confidence is low. Unlike these methods that rely on decomposing questions or generating rationales through additional models, our approach directly extracts and interprets features from pre-trained image encoders. This enables faithful and transparent multimodal reasoning without the need for extensive fine-tuning or supplementary VQG models.

**Radiological VLMs**   Several works have finetuned or otherwise trained specialized foundation models for radiological applications including Med-flamingo (Moor et al., 2023), Med-PaLM M (Tu et al., 2024), LLava-Med (Li et al., 2024), Med-Gemini (Yang et al., 2024), Rad-DINO (Pérez-García et al., 2024), MAIRA-1 (Hyland et al., 2023), R2gengpt (Wang et al., 2023b), and Radiology-GPT (Liu et al., 2023c). Regarding radiology report synthesis, many models produce both the 'findings' and 'impression' sections of the reports (Chen et al., 2020; Jin et al., 2024; Yan et al., 2023), whilst others only produce the 'impression' section (Bannur et al., 2023), or the 'findings' section (Tu et al., 2024; Miura et al., 2020; Delbrouck et al., 2022; Tanida et al., 2023; Nicolson et al., 2023). As noted by others (Hyland et al., 2023; Yu et al., 2023; Jeong et al., 2024), studies examining all three settings found that the choice of section(s) to report significantly affects the performance metrics, making comparison between results difficult. As such, we focus on the most common setting of producing the 'findings' section.

**Mechanistic interpretability**   Bricken et al. (2023) demonstrated that SAEs could recover monosemantic features by training on the residual stream of small transformers. The gated SAE was a Pareto improvement over the baseline SAE in terms of sparsity (L0) and the loss recovered (Rajamanoharan et al., 2024). There was concern that SAEs would not scale to frontier transformers until recent work by Templeton et al. (2024), which trained SAEs on Claude 3 Sonnet and discovered many monosemantic features. Similarly, Gao et al. (2024) trained SAEs on GPT-4 and introduced the *Top-K* training variant, while Lieberum et al. (2024) released Gemma Scope, a comprehensive suite of SAEs for Gemma 2 models. Contemporaneously, SAEs were trained on the class tokens of a CLIP vision transformer (Fry, 2024), InceptionV1 (Gorton, 2024), the conditioning embeddings of diffusion models (Daujotas, 2024), and the vision transformer of a pathology foundation model (Le et al., 2024). Other methods use an overcomplete basis and LASSO regressions on CLIP embeddings (Bhalla et al., 2024). However, in all cases the discovered language/visual features were not used to perform a downstream multimodal reasoning task.

## 3   BACKGROUND

In this section we give a brief overview of mechanistic interpretability, SAEs, and gated SAEs, before introducing our SAE-Rad framework.

### 3.1   MECHANISTIC INTERPRETABILITY AND SPARSE AUTOENCODERS (SAEs)

**Mechanistic interpretability**   Mechanistic interpretability research aims to identify, understand, and verify the algorithms that an ML model implements by reverse engineering a model's computations into human-interpretable components (Olah et al., 2020; Rajamanoharan et al., 2024). Classical approaches attempted to achieve this by analysing the firing patterns of individual neurons, which were interpreted as possible 'concept representations'. However, this was broadly ineffective as neurons can be *polysemantic*, meaning that a single neuron may fire on many unrelated concepts (Rajamanoharan et al., 2024; Bolukbasi et al., 2021; Elhage et al., 2022a). Polysemantic neurons are believed to arise during training due to the composition of both the linear representation and superposition hypotheses.

**Linear representation and superposition hypotheses**   Motivated by a number of findings which suggest that concept representations are linear (Gurnee et al., 2023; Olah et al., 2020; Park et al., 2023), the linear representation hypothesis states that neural networks represent concepts as directions in *activation space* (Nanda et al., 2024). This hypothesis can be thought of as being composed of two properties: 1) Linearity: That is, features are represented as directions; and 2) Decomposability: We can understand neural network outputs as a composition of multiple independently understandable features (Elhage et al., 2022b). The superposition hypothesis states that for an intermediate representation of dimension $n$, neural networks will encode $M \gg n$ concepts as linear directions (Rajamanoharan et al., 2024; Elhage et al., 2022b). These directions form an overcomplete basis of the activation space and must therefore necessarily overlap with each other. However, a single input will only activate a sparse subset of these concepts, leading to minimal interference between the (non-orthogonal) concept directions (Gurnee et al., 2023; Rajamanoharan et al., 2024). Recent work (Bricken et al., 2023) has proposed using SAEs to take features out of superposition and learn monosemantic interpretable representations.

**Sparse autoencoders (SAEs)**   SAEs attempt to 'undo' superposition by learning the sparse over-complete basis (Mallat & Zhang, 1993; Rajamanoharan et al., 2024) (or dictionary) of the activation space induced by superposition. SAEs attempt to learn both the concept directions and a sparse vector of coefficients for the inputs, that reflect how much each concept is activated for each input (Cunningham et al., 2023; Bricken et al., 2023). We will henceforth refer to such sparse vectors of coefficients as 'feature activations'.

We begin by defining the 'baseline SAE' described by Bricken et al. (2023). Let $n$ be the dimension of the input and output (typically the input is the residual stream of a transformer, and the output is its reconstruction (Elhage et al., 2021)), and $m$ be the SAE hidden layer dimension. Let $s$ be the size of the dataset. Then given encoder weights and biases $W^{\text{enc}} \in \mathbb{R}^{m \times n}, \mathbf{b}^{\text{enc}} \in \mathbb{R}^m$, and decoder weights and biases $W^{\text{dec}} \in \mathbb{R}^{n \times m}, \mathbf{b}^{\text{dec}} \in \mathbb{R}^n$, the encoding and decoding operations for a dataset $X \in \mathbb{R}^{s,n}$ are

$$\mathbf{h}(\mathbf{x}) := \text{ReLU}(W^{\text{enc}}(\mathbf{x} - \mathbf{b}^{\text{dec}}) + \mathbf{b}^{\text{enc}}) \tag{1}$$

$$\hat{\mathbf{x}}(\mathbf{h}(\mathbf{x})) := W^{\text{dec}}\mathbf{h}(\mathbf{x}) + \mathbf{b}^{\text{dec}}. \tag{2}$$

The loss function is then

$$\mathcal{L}(\mathbf{x}) := \frac{1}{|X|} \sum_{\mathbf{x} \in X} \left[ \|\mathbf{x} - \hat{\mathbf{x}}(\mathbf{h}(\mathbf{x}))\|_2^2 + \lambda \|\mathbf{h}(\mathbf{x})\|_1 \right], \tag{3}$$

where $\lambda$ is an L1 sparsity coefficient. The first term is a reconstruction error measured by the squared distance between the input and its reconstruction, and the second is an L1 regularization loss to induce sparsity. The column-wise norm of the decoder $W^{\text{dec}}$ is constrained to unit norm during training (Bricken et al., 2023).

## 3.2 GATED SAEs

As can be seen in Eq. (3), SAEs jointly optimize two opposing objectives: 1) Reconstruction fidelity and 2) L1 regularization as a proxy for sparsity (as measured by L0). This means the SAE is free to trade-off some reconstruction fidelity in order to perform better on the sparsity penalty. One consequence of this is shrinkage (Wright & Sharkey, 2024). That is, for a fixed decoder, the sparsity penalty pushes the feature activations $\mathbf{h}(x)$ towards zero whilst the squared distance loss encourages $\mathbf{h}(x)$ to be large enough in order to produce high quality reconstructions. Thus, the standard SAE will systematically underestimate the optimal magnitude of feature activations (and simply rescaling these does not necessarily overcome this bias) (Rajamanoharan et al., 2024; Wright & Sharkey, 2024).

Rajamanoharan et al. (2024) propose a gated SAE which separates the encoding procedure into two tasks: 1) Detecting which features should activate for a given input (this requires an L1 penalty if the features are to be sparse); and 2) Estimating the magnitude of the feature activations (this does **not** require an L1 loss; indeed, including this loss here introduces a shrinkage bias). The architecture of the gated encoder is

$$\mathbf{h}(\mathbf{x}) := \underbrace{\mathbb{I}[\overbrace{W^{\text{gate}}(\mathbf{x} - \mathbf{b}^{\text{dec}}) + \mathbf{b}_{\text{gate}}}^{\boldsymbol{\pi}_{\text{gate}}(\mathbf{x})} > 0]}_{\mathbf{h}_{\text{gate}}(\mathbf{x})} \odot \underbrace{\text{ReLU}(W^{\text{mag}}(\mathbf{x} - \mathbf{b}^{\text{dec}}) + \mathbf{b}_{\text{mag}})}_{\mathbf{h}_{\text{mag}}(\mathbf{x})}, \tag{4}$$

where $\mathbb{I}[\bullet > 0]$ is an element-wise Heaviside step function and $\odot$ is element-wise multiplication. The $\mathbf{h}_{\text{gate}}$ sub-function learns which features should activate for a given input and $\mathbf{h}_{\text{mag}}$ estimates the magnitude of activations for these features. Here, $\boldsymbol{\pi}_{\text{gate}}$ is referred to as the $\mathbf{h}_{\text{gate}}$ sub-function's 'pre-activations'. To minimize the number of additional parameters required, $W^{\text{mag}}$ shares the same feature directions as $W^{\text{gate}}$, and is defined as $W_{i,j}^{\text{mag}} := \exp(\mathbf{r}_i^{\text{mag}}) \cdot W_{i,j}^{\text{gate}}$, where $\mathbf{r}^{\text{mag}} \in \mathbb{R}^m$ is a vector-valued scaling parameter. Letting $\text{RA}(\cdot) := \text{ReLU}(\boldsymbol{\pi}_{\text{gate}}(\cdot))$ denote the rectified pre-activations of the gating sub-function, the loss function is defined as

$$\mathcal{L}(\mathbf{x}) := \underbrace{\|\mathbf{x} - \hat{\mathbf{x}}(\mathbf{h}(\mathbf{x}))\|_2^2}_{\mathcal{L}_{\text{reconstruct}}} + \underbrace{\lambda \|\text{RA}(\mathbf{x})\|_1}_{\mathcal{L}_{\text{sparsity}}} + \underbrace{\|\mathbf{x} - \hat{\mathbf{x}}_{\text{frozen}}(\text{RA}(\mathbf{x}))\|_2^2}_{\mathcal{L}_{\text{aux}}}, \tag{5}$$

where $\hat{\mathbf{x}}_{\text{frozen}}$ is a fixed copy of the decoder so that gradients from the auxiliary loss $\mathcal{L}_{\text{aux}}$ do not back-propagate to the decoder weights or bias terms. The auxiliary term $\mathcal{L}_{\text{aux}}$ ensures that $\mathbf{h}_{\text{gate}}$

correctly identifies features necessary for reconstruction, as its (positive) pre-activations must be able to reproduce the input. The sparsity term $\mathcal{L}_{\text{sparsity}}$ applies an L1 penalty to the rectified pre-activations (and thus sparsity is only imposed on the gating sub-function), and the reconstruction term serves the same function as in Eq. (3).

# 4   SAE-RAD

In this section we introduce SAE-Rad. First, we describe the autoencoder architecture, which is based on the gated SAE described in Section 3.2. Then, we describe our end-to-end radiology report generation pipeline.

## 4.1   SAE ARCHITECTURE

It was recently demonstrated that a lower overall SAE loss is achievable without constraining the L2 norm of the decoder weights, or centering the input based on the decoder bias (Conerly et al., 2024). Concretely, they use the same decoder as in Eq. (2) and define the encoder as

$$\mathbf{h}(\mathbf{x}) := \text{ReLU}(W^{\text{enc}}\mathbf{x} + \mathbf{b}^{\text{enc}}). \tag{6}$$

Note that the input $\mathbf{x}$ is no longer centered by subtracting the decoder bias $\mathbf{b}^{\text{dec}}$ as in Eq. (1). The sparsity penalty in the loss also now includes the L2 norm of the columns of the decoder $W^{\text{dec}}$. We present a novel proof of the equivalence between the equation below and the loss in Eq. (3) in Appendix K:

$$\mathcal{L}(\mathbf{x}) := \|\mathbf{x} - \hat{\mathbf{x}}(\mathbf{h}(\mathbf{x}))\|_2^2 + \lambda \sum_i \mathbf{h}_i(\mathbf{x}) \cdot \|W^{\text{dec}}_{\cdot,i}\|_2. \tag{7}$$

The feature activation for a feature $i$ is then $\mathbf{h}_i(\mathbf{x}) \cdot \|W^{\text{dec}}_{\cdot,i}\|_2$. The 'concept directions' are the unit-normalized decoder vectors $\frac{W^{\text{dec}}_{\cdot,i}}{\|W^{\text{dec}}_{\cdot,i}\|_2}$.

The SAE-Rad sparse autoencoder is a hybrid architecture which combines a gated encoder layer with unconstrained decoder norms. Its encoder is defined as

$$\mathbf{h}(\mathbf{x}) := \mathbb{I}[W^{\text{gate}}\mathbf{x} + \mathbf{b}^{\text{gate}} > 0] \odot \text{ReLU}(W^{\text{mag}}\mathbf{x} + \mathbf{b}^{\text{mag}}), \tag{8}$$

and the training objective given by

$$\mathcal{L}(\mathbf{x}) := \underbrace{\|\mathbf{x} - \hat{\mathbf{x}}(\mathbf{h}(\mathbf{x}))\|_2^2}_{\mathcal{L}_{\text{reconstruct}}} + \underbrace{\lambda \sum_i \text{RA}_i(\mathbf{x}) \cdot \|W^{\text{dec}}_{\cdot,i}\|_2}_{\mathcal{L}_{\text{sparsity}}} + \underbrace{\|\mathbf{x} - \hat{\mathbf{x}}(\text{RA}(\mathbf{x}))\|_2^2}_{\mathcal{L}_{\text{aux}}}. \tag{9}$$

There are four architectural differences between the gated SAE and SAE-Rad: 1) The L2 norm of the decoder weights are not constrained to unit norm; 2) The L2 norm of the decoder weights are included in the sparsity loss term; 3) The input is not centered by subtracting the decoder bias term; 4) We do not leverage stop gradient operations/decoder copying – this means that we allow gradients to back-propagate to the decoder parameters from the auxiliary loss. SAEs are trained to balance reconstruction fidelity and sparsity, placing them on a Pareto frontier of these objectives (Rajamanoharan et al., 2024). We therefore evaluate SAEs based on the L0 norm and the mean-squared error loss as defined in Eq. (3). We demonstrate that this novel SAE architecture outperforms the architecture described by Conerly et al. (2024) on our dataset, achieving both a lower L0 and a lower mean-squared error. As such, SAE-Rad is a more Pareto-optimal architecture in the present context. Details of this comparison can be found in Appendix B.1.

## 4.2   SAE-RAD – AUTOMATED RADIOLOGY REPORTING PIPELINE

In this section we describe our pipeline to automate the task of radiology report generation. Concretely, a radiographic image $\mathbf{x}$ is passed through a pre-trained and frozen vision encoder $f_{\text{img}}(\cdot) : \mathbf{x} \mapsto \mathbf{z}$ to produce an image latent $\mathbf{z}$. We leverage the hybrid SAE architecture described in Section 4.1 to learn feature directions $\frac{W^{\text{dec}}_{\cdot,i}}{\|W^{\text{dec}}_{\cdot,i}\|_2}$ and their associated activations $\mathbf{h}(\mathbf{z}) \cdot \|W^{\text{dec}}\|_2$ from the latents.

In order to generate a text-based report, we produced plain-English descriptions of the learned sparse dictionary (i.e., a description of what each feature direction represents). To do this we performed automated interpretability (Bricken et al., 2023) by using a pre-trained and frozen LLM to analyse the ground-truth radiology reports of the highest activating images for each feature. Let $\mathbf{X}_{\text{highest}}^{(i)}$ represent the set of images with the largest feature activations for feature $i$. For each image $\mathbf{x} \in \mathbf{X}_{\text{highest}}^{(i)}$ there is an associated ground-truth radiology report $r(\mathbf{x})$. We collect these reports into a set $\mathcal{R}^{(i)} := \{r(\mathbf{x})|\mathbf{x} \in \mathbf{X}_{\text{highest}}^{(i)}\}$. We then utilized a pre-trained and frozen language model $f_{\text{descriptor}} : \mathcal{R} \mapsto d$ to generate a description $d^{(i)}$ for a feature $i$ by analyzing the set $\mathcal{R}^{(i)}$ as $d^{(i)} = f_{\text{descriptor}}(\mathcal{R}^{(i)})$. This process yielded a set of feature descriptions $\{d^{(i)}\}_{i=1}^{M}$, where $M$ is the total number of features learned by the SAE.

For a new image $\mathbf{x}$ we identified the set of active features $I(\mathbf{x})$ based on a threshold $\tau$ as

$$I(\mathbf{x}) := \{i | \mathbf{h}_i(f_{\text{img}}(\mathbf{x})) \cdot \|W^{\text{dec}}\|_2 > \tau\}. \tag{10}$$

The automated radiology report $R(\mathbf{x})$ is then generated by a pre-trained and frozen LLM from the descriptions of the active features

$$R(\mathbf{x}) = f_{\text{generator}}(\{d^{(i)} | i \in I(\mathbf{x})\}). \tag{11}$$

## 5 EXPERIMENTS

Our overarching hypotheses are that: 1) SAE features capture meaningful visual concepts even in homogeneous datasets (such as is the case for chest radiographs); 2) The visual concepts captured by an SAE can be appropriately described by a pre-trained LLM by use of automated interpretability techniques with paired text data; 3) Natural language descriptions of visual features in the latent space of an SAE can be composed into high-quality radiology reports without explicit use (or training/fine-tuning) of a VLM for multimodal reasoning. We assessed these hypotheses with our **automated radiology reporting** experiment (Section 5.1). We then conducted a number of **ablation studies** to investigate the effects of model size, different sparsity constraints, and the inclusion of auxiliary information (Section 5.2). Next, we performed a case study for **image-based feature localization** to assess whether the features learned by our SAE relate appropriately to their corresponding location in the images (Section 5.3). Finally, we conducted a **reader study** with a specialist radiologist to assess the quality of our generated reports (Section 5.4).

### 5.1 AUTOMATED RADIOLOGY REPORTING

**Dataset description** We trained and evaluated all models on the MIMIC-CXR dataset (Johnson et al., 2019), a public dataset of 227,835 radiographic studies for a total of 377,110 chest radiographs and associated written text reports. We linked all images to their DICOM metadata files to retrieve scan orientations. We only considered images for which metadata files exist, and retained only antero-posterior(AP)/postero-anterior(PA) scans for training (these are 'head-on' scans, as opposed to lateral ones). These are the default views for the generation of diagnostic reports given the increased clarity and ability to visualize the relevant anatomy comprehensively (Hyland et al., 2023). We then extracted the 'findings' section from each text report. The 'findings' are a natural language description of all relevant negative and positive features for a given radiograph. Occasionally, the 'findings' section is placed into another section with the sub-heading of 'impression'. Datapoints without either a findings or impression section were discarded. We used the recommended train/test split for MIMIC-CXR, resulting in a total of 239,931 training and 3,403 test images.

**Evaluation metrics** We evaluated generated radiology reports using both general Natural Language Generation (NLG) metrics (BLEU-4 (Papineni et al., 2002), ROUGE-L (Lin, 2004), METEOR (Banerjee & Lavie, 2005)) and radiology-specific metrics (RGER score (Delbrouck et al., 2022), CheXpert F1 score (Irvin et al., 2019)). While lexical metrics assess n-gram overlap and word order, clinical metrics like RGER and CheXpert F1 attempt to evaluate factual completeness and consistency by analyzing entity-relationship graphs and predicting common chest X-ray pathologies, respectively. NLG metrics can be inadequate for assessing radiology reports as they don't account for clinical significance (Bannur et al., 2024), while radiology-specific metrics often

rely on specialized models (Yu et al., 2023) or pre-specified findings classes (Smit et al., 2020; Bannur et al., 2024). To address these limitations, the RadFact framework Bannur et al. (2024) uses LLMs to assess sentence-level factuality through bi-directional entailment verification with reference reports— offering a robust evaluation method without relying on pre-specified error types or specialized models. Additional details are provided in Appendix B.2.

**Experimental setup** The SAE-Rad framework was trained on class tokens produced by the Rad-DINO vision transformer, using an expansion factor of 64 resulting in a latent dimension of 49,152. The model was optimized using Adam with a learning rate peaking at 5e-5 and a sparsity penalty of 8e-3, trained for 200,000 steps with a batch size of 2048. Claude 3.5 Sonnet was used for automated feature interpretation and report generation, while RadFact evaluation employed Llama3-70B-Instruct. Additional experimental setup details are given in Appendix B.3.

Table 1: **Report generation performance on the official MIMIC-CXR test split.** BL4 = BLEU-4, RG-L = ROUGE-L, MTR = Meteor. Ma-5 (Macro-F1-5), Ma-14 (Macro-F1-14), Mi-5 (Micro-F1-5), and Mi-14 (Micro-F1-14) represent the clinical CheXbert labeler scores. Bolding represents best performance in the current study or between the upper bound models.

| Model | RadFact ↑ | NLG Metrics ↑ | | | Clinical Metrics ↑ | | | | |
|---|---|---|---|---|---|---|---|---|---|
| | F1 | BL4 | RG-L | MTR | RGER | Ma-5 | Ma-14 | Mi-5 | Mi-14 |
| *Current study* | | | | | | | | | |
| Baseline | 30.0 | 3.1 | 18.6 | 23.5 | 14.9 | 38.3 | 25.5 | 45.0 | 42.8 |
| CheXagent | 36.9 | **3.7** | **21.5** | 21.1 | 18.0 | 31.5 | 22.5 | 38.6 | 38.1 |
| SAE-Rad (×64) | **37.2** | 1.9 | 17.1 | **29.1** | **18.2** | **47.2** | **34.3** | **54.4** | **53.2** |
| *Upper bound models* | | | | | | | | | |
| MAIRA-1 | 47.8 | 14.2 | 28.9 | 33.3 | 29.6 | 47.7 | 38.6 | 56.0 | 55.7 |
| MAIRA-2 | **50.4** | **23.1** | **38.4** | **41.7** | **39.6** | **50.4** | **41.6** | **59.1** | **58.1** |

Feature 1 DIMENSION 714
*DEXTROSCOLIOSIS OF THE THORACIC SPINE*

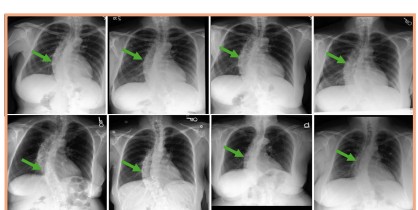

Feature 3 DIMENSION 150
*RIGHT-SIDED PLEURAL EFFUSION*

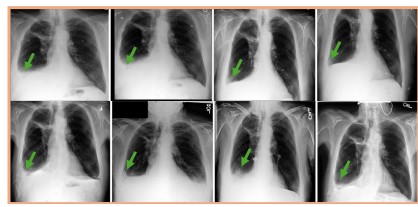

Feature 2 DIMENSION 152
*BILATERAL LUNG OPACITIES & SIGNS OF PULMONARY CONSOLIDATION*

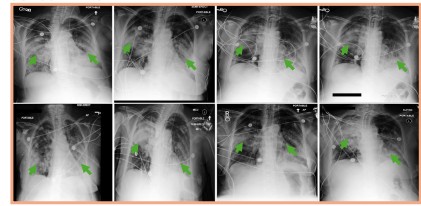

Feature 4 DIMENSION 89
*PACEMAKER DEVICE & ASSOCIATED LEADS WITHIN CHEST CAVITY*

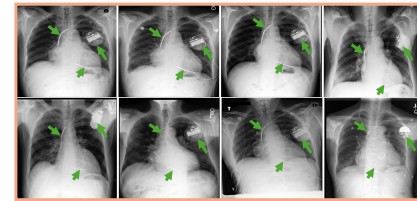

Figure 2: **SAE-Rad identifies clinically relevant and interpretable features within radiological images.** We illustrate a number of pathological and instrumentation features relevant for producing radiology reports. We add annotations (green arrows) to emphasize the presence of each feature.

**Quantitative evaluation** We compared SAE-Rad to the current state-of-the-art radiology reporting systems. CheXagent (Chen et al., 2024) is an instruction-tuned foundation model for CXRs trained on 1.1M scans for question-answering and text-generation tasks. MAIRA-1 &-2 (Hyland et al., 2023; Bannur et al., 2024) are VLMS based on the LLaVA 1.5 architecture (Liu et al., 2024b;a). MAIRA-2 is trained on 510,848 CXRs from four datasets and sets the current state-of-the-art for report generation. The MAIRA systems are not publicly available for result replication, and

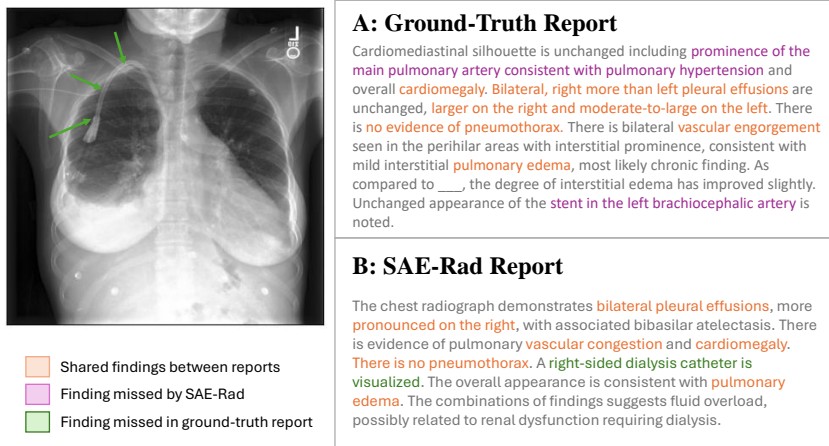

Figure 3: **SAE-Rad accurately captures features reported by human radiologists and more**. Above, we showcase a side-by-side comparison between a ground-truth radiology report and one generated by SAE-Rad. The model successfully identifies key clinically relevant features. SAE-Rad also identifies additional details, such as a right-sided dialysis catheter, without hallucination (we annotate this feature with green arrows for emphasis). SAE-Rad can also miss features when compared to the reference report.

thus we quote their evaluation values directly as our upper-bound. CheXagent is publicly available, and we therefore performed independent replications for this model for a direct comparison. The 'baseline' approach is a naïve method of report generation that uses the report of the closest image in the MIMIC train split. Further details of the 'baseline' approach can be found in Appendix B.6. As Table 1 demonstrates, SAE-Rad underperforms on generic NLG metrics such as BLEU-4. This is expected as we do not try to optimize for any specific 'writing style' by fine-tuning an LLM on the reference reports from MIMIC-CXR. Conversely, SAE-Rad demonstrates strong performance on radiology-specific metrics which are clinically relevant, outperforming CheXagent by up to 52% in the CheXpert F1 score (macro-averaged F1-14), and achieving 92.1% and 89.9% of the performance of MAIRA-1 and MAIRA-2 on these scores, respectively. We additionally compare SAE-Rad to 14 alternative approaches for automated report generation in Appendix J, broadly outperforming them across all clinical metrics.

**Qualitative investigation** Figure 2 illustrates randomly selected monosemantic visual features from SAE-Rad. As can be seen, the SAE learns human-interpretable visual concepts despite the homogeneity and relatively small size of the dataset. These include dextroscoliosis of the spine (Fig. 2; feature 1), bilateral opacifications (Fig. 2; feature 2), unilateral pleural effusions (Fig. 2; feature 3), and the presence of instrumentation – in this case a pacemaker (Fig. 2; feature 4). In Fig. 3, we illustrate an example 'findings' section for a CXR with a number of pathological findings; SAE-Rad is capable of detecting multiple relevant pathologies for a given image. Like other radiology report generation systems, SAE-Rad can miss findings. However, it can also occasionally describe a relevant finding which is otherwise missing from the reference report – an example relating to the presence of a dialysis catheter is shown in Fig. 3.

## 5.2 ABLATION STUDIES

We conducted a set of additional experiments which characterize the effects of: 1) Varying the SAE expansion factor; 2) Investigating less sparse ('dense') SAEs by reducing the L1 penalty coefficient; and 3) Evaluating the relative benefits of including auxiliary information including the 'indication' section of a report as well as previous reports, where available. The experimental setup for these ablations is described in Appendix H.

As can be seen in Table 2, an expansion factor of ×64 produced a higher RadFact F1 score compared with both smaller (×32) and larger (×128) expansion factors. In addition, denser SAEs

with a larger L0 norm underperformed sparser models. This suggests that concepts useful for radiology report generation are likely to exist in balance between being too coarse or too fine-grained. If the feature are too coarse, this may cause 'feature absorption'— an asymmetric form of feature splitting that can negatively impact the interpretability of an SAE feature (Chanin et al., 2024). Conversely if the features are too fine-grained, the features may be more difficult to accurately describe given insufficient amount of descriptive detail in the ground-truth reference reports.

Table 2 demonstrates that the addition of auxiliary information such as the indication, which describes why the patient required the scan in the first instance, can boost the RadFact F1 score, with a large boost to recall. However, in our experiments this caused a small degradation to the precision sub-metric. This finding runs against prior work on the effect of including this section (Yu et al., 2023; Tu et al., 2024; Nguyen et al., 2023) and warrants additional investigation in future. We find that adding both previous indications and prior studies has a net positive effect on the quality of generated reports. Additionally, we report NLG and radiology-specific metrics for this ablation experiment in Table 6 of Appendix H, These additional analyses support the findings in Table 2.

Table 2: RadFact performance metrics for different SAE-Rad configurations. /w inds = with indication(s), /w inds + prev.reps = with indications and previous text reports.

| SAE-Rad Configuration | Precision ↑ | Recall ↑ | F1 Score ↑ |
|---|---|---|---|
| ×128 | 34.83 | 29.91 | 32.18 |
| ×64 | **35.95** | **31.95** | **33.83** |
| ×32 | 31.22 | 27.89 | 29.46 |
| ×128_dense | 32.57 | 27.06 | 29.56 |
| ×64_dense | 31.15 | 28.46 | 29.74 |
| ×32_dense | 32.02 | 28.61 | 30.22 |
| ×64 /wo inds | **38.78** | 28.22 | 32.67 |
| ×64 /w inds | 38.45 | 32.42 | 35.18 |
| ×64 /w inds + prev. reps | 37.32 | **39.83** | **38.45** |

### 5.3 IMAGE-BASED FEATURE LOCALIZATION

To test whether SAE features relate appropriately to their corresponding image location we trained a diffusion model conditioned on Rad-DINO class tokens (Pérez-García et al., 2024). We then produced *counterfactual* images, where we ask, what would this image have looked like if a particular feature were present or absent? If the SAE successfully captures visual features, the counterfactual images should reflect the targeted feature's presence or absence. To produce such images, we passed a class token through the SAE, intervened on encoder activations, and reconstructed a counterfactual token via the decoder, which conditioned the diffusion model to project interventions into imaging space. We tested whether: 1) interventions alter the reconstructed class token accordingly, 2) changes affect only the targeted feature, and 3) features can be "added" or "removed" by manipulating the same activation. Fig. 4 shows the results for two features (cardiomegaly and pacemaker), demonstrating that our interpretations accurately reflect their impact on model behaviour. This also enables us to conduct *unsupervised segmentation*, where we semantically segment specific visual features by adding them to an image and then taking the difference between the pre- and post-edit images. Further details are given in Appendix D.

### 5.4 READER STUDY

In a reader study with a specialist radiologist evaluating the quality of the automated radiology reports, 328 sentences from 60 reports (SAE-Rad, CheXagent, and a baseline) were analyzed. SAE-Rad had up to 7% fewer edits than other models and demonstrated significantly fewer errors with clinical impact, particularly in the "significant" category, where SAE-Rad had almost half the rate compared to others. This highlights SAE-Rad's potential for radiology report generation in a real clinical scenario. The full study can be found in Appendix E.

### 6 DISCUSSION, LIMITATIONS, AND CONCLUSION

In this work, we introduced SAE-Rad, a novel framework that leverages sparse autoencoders to automate radiology report generation. Our approach directly decomposes image class tokens from a pre-trained radiology image encoder into human-interpretable features, which are then compiled into comprehensive radiology reports. The experimental results demonstrate that SAE-Rad achieves competitive performance on radiology-specific metrics, outperforming existing models like CheXagent (Chen et al., 2024) and approaching the performance of state-of-the-art systems such as

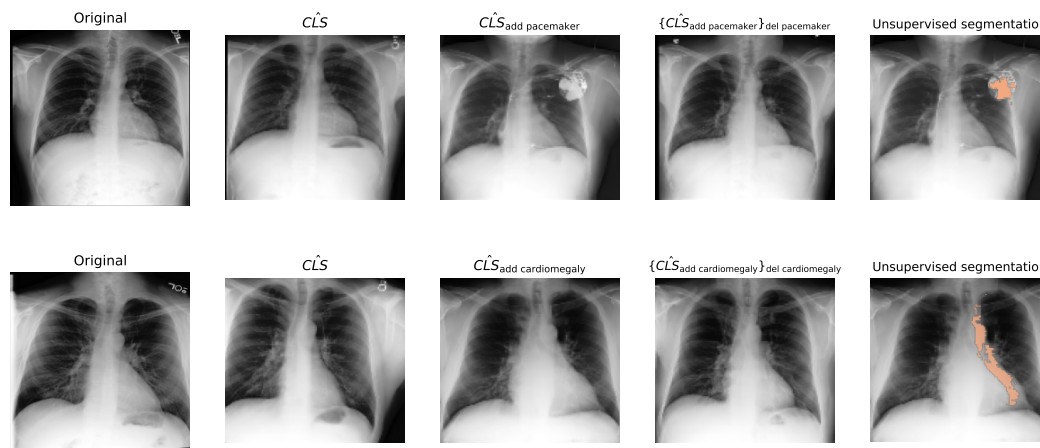

Figure 4: **SAE-Rad enables targeted counterfactual image generation and unsupervised segmentation with disentangled class tokens.** Row 1 examines a pacemaker, and Row 2 investigates cardiomegaly. Column 1 shows original MIMIC-CXR images, Column 2 shows model reconstructions, and Columns 3 and 4 depict counterfactuals by adding and removing features. The final column demonstrates unsupervised segmentation by comparing counterfactual and original images. Details are in Appendix D.

MAIRA-2 whilst being trained on a significantly smaller dataset and with a much lower training compute budget; see Appendix F. By reverse-engineering the computations of the image encoder, SAE-Rad provides a framework that is verifiably faithful to the underlying model, enhancing transparency and trustworthiness, which are critical considerations in the healthcare setting.

Qualitative analyses confirm that SAE-Rad successfully captures meaningful visual concepts including the presence or absence of pathological features. These interpretable features contribute to generating detailed and accurate radiology reports, as evidenced by strong performance in the clinical evaluation metrics. Our ablation studies indicate that the choice of expansion factor and inclusion of auxiliary information, such as previous reports and indications, can significantly impact the quality of the generated reports, which is broadly in line with the previous literature on this topic (Bannur et al., 2024).

Our approach has limitations. First, SAE-Rad relies on pre-trained (frozen) models for both the image encoder and the LLM in the interpretability pipeline, potentially introducing inherent biases. Spurious correlations and other biases in the image encoder's latent space may prevent our method from fully disentangling concepts. This may lead to *feature absorption* (Chanin et al., 2024), where distinct concepts are inappropriately merged into a single SAE dimension. However, due to the pipeline's modular nature, these biases can be mitigated by replacing either model without retraining the SAE if the LLM is swapped. Additionally, SAE-Rad underperforms on general language metrics like BLEU-4, suggesting that while the generated reports are clinically accurate, they may lack the fluency and stylistic nuances of human-generated reports, particularly those in the MIMIC-CXR dataset. Improving these metrics through style-aware radiology report generation (Yan et al., 2023) is a natural avenue for future work. Another exciting direction is to prevent errors from LLM hallucinations by having SAE features labeled by human inspection (Appendix I) and using a deterministic regex for report synthesis, thereby obviating the need for LLMs in our pipeline altogether.

Overall, this feasibility study presents a novel and effective approach to radiology report generation by leveraging mechanistic interpretability techniques to extract and utilize human-interpretable features from medical images. By providing a verifiably faithful representation of the underlying computations, SAE-Rad contributes to the development of more interpretable AI systems in healthcare.

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

# Appendix

# Table of Contents

## A    REPRODUCIBILITY STATEMENT

We provide details on the experimental setup, hyperparameters, and data preprocessing steps to ensure the reproducibility of our results. We cannot share the MIMIC-CXR dataset, however it can be accessed through application here: `https://physionet.org/content/mimic-cxr/2.1.0/`. Our codebase is available at: `https://anonymous.4open.science/r/sae_rad-5B77/`.

## B    AUTOMATED RADIOLOGY REPORT EXPERIMENT

### B.1    SAE ARCHITECTURE COMPARISON

In this section we compare the performance of our novel SAE architecture with the state-of-the-art architecture proposed by Conerly et al. (2024), which itself introduced a novel loss that outperformed prior works. The following hyperparameters were used to train both SAEs:

- Expansion factor of $\times 64$.

- Batch size of 2048.

- Learning rate of $5 \times 10^{-5}$.

- Linear warm-up of learning rate for the first 1% of training.

- Linear warm-down of learning rate for the last 20% of training.

- L1 coefficient warmup for the first 5% of training.

- Adam optimizer with no weight decay.

- Trained for 200,000 optimization steps.

The L1 coefficient was increased for the SAE-Rad architecture in comparison to the Conerly et al. (2024) SAE in order to compensate for the additional auxiliary loss term used to train the SAE-Rad architecture. Table 3 displays the resulting metrics comparing the two SAEs - our proposed SAE attains both a lower L0 and a higher reconstruction accuracy.

|  | L1 coefficient | L0 $\downarrow$ | Explained Variance (%) $\uparrow$ |
|---|---|---|---|
| Conerly et al. (2024) SAE | $5.3 \times 10^{-3}$ | 15.7 | 81.3 |
| SAE-Rad | $8.0 \times 10^{-3}$ | **13.6** | **84.3** |

Table 3: Comparison of SAE types.

### B.2    EVALUATION METRICS

**NLG and classical radiology-specific metrics**    We evaluated generated radiology reports using both general NLG metrics and radiology-specific metrics. For lexical evaluation, we report BLEU-4 (Papineni et al., 2002) for 4-gram overlap based on n-gram precision, ROUGE-L (Lin, 2004) for longest common subsequence matching, and METEOR (Banerjee & Lavie, 2005), which performs unigram matching using surface forms, stems, and meanings, computing scores based on precision, recall, and fragmentation assessing word order. Whilst widely reported, lexical metrics do not capture factual completeness or consistency (Miura et al., 2020; Bannur et al., 2024) and we therefore also include classical clinical metrics. The RGER score (Delbrouck et al., 2022), based on the RadGraph model (Jain et al., 2021), evaluates entity-relationship graphs extracted from reports by matching entities and verifying relationships. Additionally, we report the CheXpert F1 score (Irvin et al., 2019), utilizing the CheXbert model (Smit et al., 2020) to predict 14 common pathologies in chest X-rays and calculating the harmonic mean of precision and recall between generated and reference texts. Following recommendations (Miura et al., 2020; Tu et al., 2024; Hyland et al., 2023), we provide micro- and macro-averaged scores over five key observations—atelectasis, cardiomegaly, consolidation, edema, and pleural effusion—as well as the F1-14 score encompassing all observations.

**RadFact — A robust evaluation framework for radiology**   Natural language generation (NLG) metrics are insufficient to appropriately assess radiology report generations as they do not account for the clinical significance of each sentence, and instead treat all words equally (Bannur et al., 2024). Conversely, radiology-specific metrics are often based on specialised models such as CheXbert (Smit et al., 2020; Irvin et al., 2019) or RadGraph (Yu et al., 2023; Jain et al., 2021; Delbrouck et al., 2022), which are themselves limited in that rely on specialised models (Yu et al., 2023), pre-specified findings classes (Smit et al., 2020; Bannur et al., 2024), and/or error types (Chaves et al., 2024; Wang et al., 2024). To this end, Bannur et al. (2024) proposed the RadFact framework which leverages LLMs to assess the factuality of *each sentence* by use of bi-directional entailment verification with the ground-truth reference report. RadFact does not rely on pre-specified error types or radiology-specialized models and is therefore significantly more robust for use to assess the quality of generated radiology reports.

### B.3   SAE-Rad ADDITIONAL EXPERIMENTAL SETUP DETAILS

In our instantiation of the SAE-Rad framework (described in Sections 4.1 and 4.2), all bias terms $\mathbf{b}^{\text{enc}}, \mathbf{b}^{\text{gate}}$, and $\mathbf{b}^{\text{mag}}$ were initialized to zeros. The elements of the shared encoder weights matrix $W^{\text{gate}}$ were initialized such that all rows point in random directions. The decoder weights $W^{\text{dec}}$ were initialized to $W^{\text{gate}\top}$. For a dataset of size $S$, we trained our SAE on the class tokens $\mathbf{Z} \in \mathbb{R}^{S \times 768}$ produced by the Rad-DINO vision transformer (Pérez-García et al., 2024), a fine-tuned Dino V2 (Oquab et al., 2023) on a large dataset of chest X-rays. We use the model weights released at `https://huggingface.co/microsoft/rad-dino`. The dataset was shuffled and scaled by a constant such that $\mathbb{E}_{\mathbf{z} \in \mathbf{Z}}[\|\mathbf{z}\|_2] = \sqrt{\dim(\mathbf{z})}$, where $\dim(\mathbf{z}) = 768$. We used an expansion factor of 64 and thus the latent dimension of our SAE is $49,152$. We used the Adam optimization algorithm (Kingma, 2014) with $\beta_1 = 0.9$, $\beta_2 = 0.999$, and no weight decay. Our learning rate was increased linearly over the first 1% of training to $5 \times 10^{-5}$ and then decayed linearly to zero over the final 20% of training. Our sparsity penalty $\lambda$ was linearly increased over the first 5% of training to $8 \times 10^{-3}$. We trained our model for $200,000$ steps. We used a batch size of 2048. We did not use ghost gradients or resampling strategies as none of our features were dead at the end of training — in other words, all learned concept directions were activated at least once when passing the dataset through the SAE. We trained all models on a single NVIDIA A6000 GPU. We used Claude 3.5 Sonnet (Anthropic, 2024) for our automated interpretability pipeline; for a given SAE feature, we retrieved the 10 highest activating images for that feature and passed their ground-truth reference report to Claude with the task of extracting the most consistent information across the reports. The output description was used to label the feature. The full prompts are described in Appendix B.4.

To generate a report from SAE features, we accrued all activating features and their descriptions, and passed them to Claude for concatenation into a full 'findings' paragraph. The full prompts are describe in Appendix B.5. We evaluate RadFact using Llama3-70B-Instruct.

### B.4   PROMPT USED TO ANNOTATE SAE FEATURES

In this section we describe the prompt used to instruct Claude 3.5 Sonnet (Anthropic, 2024) to annotate SAE features. Overall, we instruct Claude to identify commonalities between radiological reports corresponding to the 10 maximally activating examples for a given SAE feature. The prompt encourages the model to use chain-of-thought (Wei et al., 2022) reasoning in-order to ensure the quality of the feature annotations.

```
1  You are an expert radiologist specializing in chest radiographs. We're
       studying neurons in an image neural network, where each neuron
       detects specific features in chest X-rays. I've identified the
       radiology images that most strongly activate a particular neuron and
       will provide you with their associated text radiology reports. Your
       task is to analyze these reports and determine the common feature
       that this neuron is detecting.
2  To arrive at the most accurate and precise explanation of what this
       neuron is detecting, you must engage in explicit chain of thought
       reasoning. Begin by thoroughly examining all provided radiology
       reports, noting any patterns or commonalities. Pay close attention to
        recurring terminology, described anatomical structures, and
       consistent pathological findings. Consider how these elements might
```

```
        interrelate to form a singular, distinctive feature that the neuron
        could be identifying. Evaluate the context of chest radiographs and
        consider which aspects would be most significant or unique within
        this imaging modality.
3  As you progress through your analysis, verbalize your thought process.
        Explain each step of your reasoning, from initial observations to
        intermediate conclusions, and finally to your overall assessment.
        This chain of thought approach will help ensure a comprehensive and
        well-reasoned final explanation.
4  After this detailed analytical process, formulate a single, specific
        explanation of what the neuron is detecting. Your explanation should
        be as precise and fine-grained as possible, avoiding vague or general
         statements. Focus on specific features or combinations of features,
        using 'and' to connect multiple elements if necessary. Avoid using '
        or' to list multiple possibilities. Refrain from explaining the
        pathology itself (e.g., avoid statements like "This feature
        represents X, which is characterized by..."). Base your explanation
        solely on the information provided in the reports, without additional
         medical knowledge that might not be captured by the neuron.
5  It is crucial that you present your final explanation in the following
        format:
6  *This feature represents [your specific, detailed description of what the
         neuron is detecting].
7  The asterisk is absolutely essential. Your explanation must begin
        immediately after the asterisk, without any additional text,
        numbering, or preamble. The presence of this asterisk is critical for
         the proper processing of your response.
8  Below are the radiology reports, listed in order of how strongly they
        activate the neuron. Use these to inform your analysis and final
        explanation:
9  Report number 1: ...
10 Report number 2: ...
11 ...
12 Report number 10: .
```

Below is an example output from Claude in response to the above prompt.

```
1  Thank you for providing these detailed radiology reports. I'll analyze
        them to determine the common feature that this neuron is detecting.
        Let me walk you through my thought process:
2
3  1. First, I'll review each report, noting recurring elements:
4     - Report 1: Mentions cardiomegaly and atherosclerotic calcifications
         in the thoracic aorta.
5     - Report 2: Notes mild cardiomegaly and a moderately tortuous, mildly
         calcified aorta.
6     - Report 3: Describes an enlarged cardiomediastinal silhouette and a
         calcified, unfolded aorta.
7     - Report 4: Mentions moderate enlargement of the cardiac silhouette.
8     - Report 5: No specific mention of heart or aorta abnormalities.
9     - Report 6: Notes moderate cardiomegaly.
10    - Report 7: Mentions moderately enlarged cardiac silhouette.
11    - Report 8: Describes moderate cardiomegaly and moderate tortuosity of
          the thoracic aorta.
12    - Report 9: No specific mention of heart or aorta abnormalities.
13    - Report 10: Notes an enlarged cardiac silhouette and a tortuous
         descending thoracic aorta.
14
15 2. Analyzing these observations, I notice two recurring themes:
16    a) Enlarged cardiac silhouette or cardiomegaly
17    b) Abnormalities of the thoracic aorta (tortuosity, calcification,
         unfolding)
18
19 3. Considering the context of chest radiographs, both of these features
        would be significant and distinctive within this imaging modality.
```

```
20
21  4. The frequency of these observations is notable:
22     - Cardiomegaly or enlarged cardiac silhouette is mentioned in 7 out of
           10 reports.
23     - Aortic abnormalities are mentioned in 5 out of 10 reports.
24
25  5. Given that cardiomegaly is mentioned more frequently, it might be the
       primary feature the neuron is detecting. However, the aortic
       abnormalities are often mentioned in conjunction with cardiomegaly.
26
27  6. It's possible that the neuron is detecting a combination of these
       features, as they often co-occur and are related to cardiovascular
       changes.
28
29  7. The neuron might be sensitive to the overall appearance of an enlarged
        cardiac silhouette combined with changes in the adjacent thoracic
       aorta, which together create a distinctive pattern on chest
       radiographs.
30
31  Based on this analysis, I believe the most precise and specific
       explanation for what this neuron is detecting is:
32
33  *This feature represents an enlarged cardiac silhouette (cardiomegaly) in
        conjunction with thoracic aortic abnormalities, particularly
       tortuosity and calcification.
```

Finally, we use regex to parse the description using the asterisk as a separator.

### B.5 PROMPT USED TO COMPOSE FINDING SECTIONS VIA SAE FEATURES

In this section we describe the prompt used to instruct Claude to compose the finding sections of a radiology report based on: 1) a patient's past radiology reports, if they exist, 2) the features that SAE-Rad has identified are present within the image and 3) the indication for the radiograph. Each feature is associated to an *importance score* in accordance to the magnitude of the SAE dimension of provenance — the higher the magnitude, the higher the importance score. The scores were normalised to the interval between 0 and 1. Our prompt encourages Claude to focus on features with the highest scores as these were likely to be more significant. See Appendix C.1 for more example outputs following this prompt, alongside their corresponding radiographs.

```
1  You are an expert radiologist specializing in chest radiographs. Your
       task is to write the findings section for a radiology report based on
        a chest X-ray image. To assist you, I may provide up to three of the
        patient's past radiology reports, if available. These might contain
       useful information related to the features of the current scan. I
       will also give you the indication (reason) for the current X-ray.
       Additionally, you'll receive text descriptions of features present in
        the current X-ray image, along with importance scores for each
       feature. Your primary focus should be on producing the findings
       section for the latest scan, given the features about that scan.
       Focus on features with higher importance scores, as these are more
       prominent in the image and should be emphasized. Assess the current
       features, and then judge whether it would be appropriate to relate
       them to information in previous scans, if provided. Do not explicitly
        mention dates and times from previous reports. Discuss the features
       present in the X-ray, along with their implications and any
       deductions you can make. Your response should constitute the '
       findings' section of the radiology report, providing a comprehensive
       analysis of the current X-ray. All of the information is provided
       below:
2  <patient_history>
3  <past_report>
4  Report number 1. This report was written 0 years, 2 days, 16 hours and 28
        minutes before the current chest x-ray
```

```
 5  INDICATION:  Left-sided pleuritic chest pain ...
 6  COMPARISON:  Chest radiograph ___ and chest CT ___ ...
 7  IMPRESSION: Ill-defined patchy opacities in lung bases which may
        represent ...
 8  </past_report>
 9  </patient_history>
10
11  <current_chest_x_ray>
12  <feature 1>
13  Feature number 1. Relative importance score 1.0:
14  This feature represents the absence of pneumothorax, characterized by
        normal lung appearance at the pleural margins.
15  </feature 1>
16  ...
17  <feature n>
18  </current_chest_x_ray>
19  Using the information provided, compose the findings section of the
        radiology report. Be aware that some of the described features may be
         inaccurate or only loosely related to the actual characteristics
        present in the X-ray. When faced with conflicting information, rely
        on the importance scores or a majority consensus to determine which
        features are most likely correct. In your report, refrain from simply
         listing the features. Avoid using the word 'feature' entirely in
        your report. Keep the radiology report brief and to the point. The
        reason for the current x-ray examination is provided below:
20
21  <indication>
22  Status post CABG.
23  </indication>
24
25  Now write the findings section. This should be a single contiguous
        paragraph with the findings of the X-ray radiology report. No more
        than 5 to 6 sentences. Be concise and avoid simply listing the
        features. Do not respond with any additional text other than the
        findings. Do not add any concluding statements at the end, only
        include findings.
```

### B.6 BASELINE EXPERIMENTAL SETUP

In this section, we provide details on the experimental details of our baseline report generation method. In summary, the baseline approach returns the 'closest' report in the train split. Explicitly, we encode a given radiograph using Rad-DINO into a single embedding vector $\mathbf{z} \in \mathbb{R}^{768}$ by selecting the class token. Then, we identify the radiograph $y$ with the closest Rad-DINO class token from the MIMIC-CXR train split $X$ as measured by the Frobenius norm $y = \arg\min_{p \in X} ||\mathbf{z}(x) - \mathbf{z}(p)||_2$. We then return the original findings section from $y$ as our predicted, baseline report.

## C ADDITIONAL QUALITATIVE EVALUATIONS

### C.1 ADDITIONAL RADIOLOGY REPORT GENERATIONS

In this section we provide additional qualitative examples of SAE-Rad generated findings sections, highlighting findings jointly described between radiologist generated reports, missing findings by SAE-Rad, and correct findings highlighted by SAE-Rad not present in the original report. Descriptions and analyses of findings (dis)similarity were produced and reviewed by a clinician.

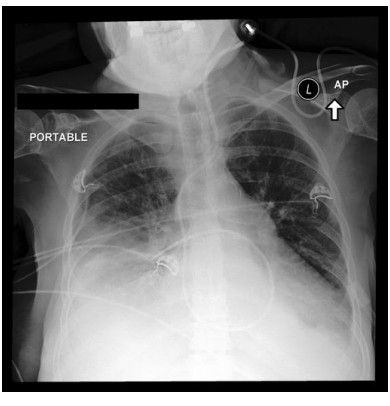

**A: Ground-Truth Report**

A left IJ Swan-Ganz catheter is present. As before, the tip lies relatively distal in the right pulmonary artery. Clinical correlation to assess for possible retraction is requested. A mesh-like stent extends along the expected course of the left subclavian and brachiocephalic vessels. Compared with the earlier film, there may have been slight interval improvement in the upper zone redistribution and vascular engorgement. Otherwise, no gross interval change is detected. Again seen is mild vascular prominence and opacity at the right base which is likely a combination of pleural fluid, possible elevated hemidiaphragm, and underlying collapse and/or consolidation. There is also increased retrocardiac density, consistent with left lower lobe collapse and/or consolidation and a small left effusion. The left effusion is slightly more apparent on the current exam. No pneumothorax is detected.

**B: SAE-Rad Report**

A Swan-Ganz catheter is visualized in the pulmonary artery, with its tip positioned relatively distally in the right pulmonary system. Bilateral basal opacities are present, more pronounced on the right side, consistent with pleural effusions and underlying atelectasis. A moderate to large right-sided pleural effusion is noted, contributing to the overall asymmetry in lung appearance. The left lung demonstrates less extensive effusion, but some fluid accumulation is still evident. Despite these findings, there is no evidence of pneumothorax. The cardiomediastinal silhouette appears unchanged compared to prior studies, and a vascular stent in the left subclavian artery remains in stable position.

Shared findings between reports

Finding missed by SAE-Rad

Finding missed in ground-truth report

Figure 5: **Comparison between SAE-Rad Report and the Ground Truth (1)**. Both reports identify a Swan-Ganz catheter terminating distally in the right pulmonary artery, a left subclavian stent, left as well as right (more pronounced) pleural effusion, and no evidence of pneumothorax. SAE-Rad correctly notes bilateral basal opacities, an asymmetry in lung appearance, consistency in the cardiomediastinal silhouette with respect to previous studies, and a moderate to large right pleural effusion. SAE-Rad misses that the stent also extends to the branchiocephalic vessels, and that that the left effusion is slightly more apparent compared to previous exams.

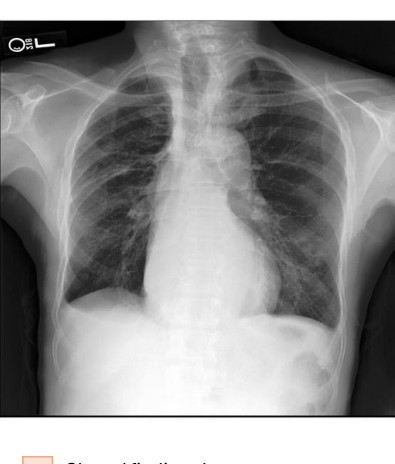

## A: Ground-Truth Report

The lungs are hyperinflated but clear of consolidation. Linear opacity in the right mid to upper lung is compatible with scarring as well as changes of the posterior right ribs which are chronic. Blunting of the right lateral posterior costophrenic angle is chronic, potentially due to scarring or trace effusion. Blunting of the left posterior costophrenic angle suggests small pleural effusion. Cardiomediastinal silhouette is within normal limits. Prominent retrocardiac opacity on the right is compatible with a neo esophagus. No acute osseous abnormalities.

## B: SAE-Rad Report

The chest radiograph demonstrates linear opacities in the right lung, consistent with scarring or subsegmental atelectasis. There are focal areas of consolidation or atelectasis noted, which may represent residual changes from previous infections or aspiration events. The right posterior pleural sulcus appears blunted, possibly due to a small chronic pleural effusion or scarring, as noted in prior studies. The cardiac silhouette is within normal limits, and there is no evidence of acute pulmonary edema or pneumothorax. Emphysematous changes are present, reflecting the patient's known history of emphysema. Post-surgical changes from previous esophagectomy and gastric pull-through are noted and unchanged from prior examinations.

Shared findings between reports

Finding missed by SAE-Rad

Finding missed in ground-truth report

Figure 6: **Comparison between SAE-Rad Report and the Ground Truth (2)**. Hyperinflation in the lungs, right lung linear opacities, blunting of the right posterior pleural sulcus and the normal limits of the cardiac sihloutee are noted in both reports. SAE-Rad detects the presence of focal areas of consolidation or atelectasis, the patients known history of emphysema, and the gastric pull-through that remains unchanged from prior exams. SAE-Rad misses the changes of the posterior right ribs and the lack of acute osseous abnormalities.

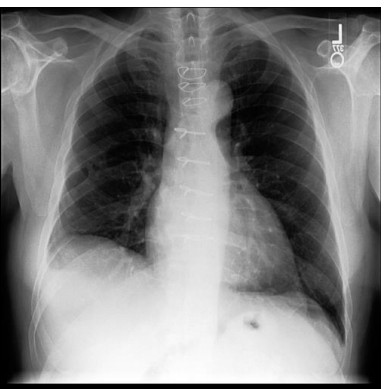

## A: Ground-Truth Report

The patient is after median sternotomy and CABG with stable appearance of heart and mediastinum. Lungs are essentially clear with no pleural effusion or pneumothorax. Elevated right hemidiaphragm is unchanged.

## B: SAE-Rad Report

The chest radiograph demonstrates normal cardiomediastinal contours and clear lung fields without evidence of focal consolidation, pleural effusion, or pneumothorax. There are post-surgical changes evident, including median sternotomy wires and associated mediastinal clips, indicative of prior cardiac surgery. The cardiac silhouette appears normal in size. The right hemidiaphragm is mildly elevated with a slight reduction in right lung volume, which may be due to the patient's positioning or represent a chronic change. No acute cardiopulmonary abnormalities are identified. Overall, the examination reveals post-surgical changes without any acute findings in the context of the patient's history of melanoma.

Shared findings between reports

Finding missed by SAE-Rad

Finding missed in ground-truth report

Figure 7: **Comparison between SAE-Rad Report and the Ground Truth (3)**. Ground-truth and SAE-Rad reports identify the normal catdiomediastinal contours, clear lung fields, as well as the lack of pleural effusion or pneumothorax. SAE-Rad misses that the patient underwent CABG and that their heart, mediastinum and elevated right hemidiaphraghm remain unchanged. SAE-Rad correctly comments on the lack of focal consolidation and cardiopulmonary abnormalities, and notes acute findings with respect to the patients past diagnosis of melanoma. Additionally, SAE-Rad notes the presence of mediastinal clips and a slight reduction in right lung volume.

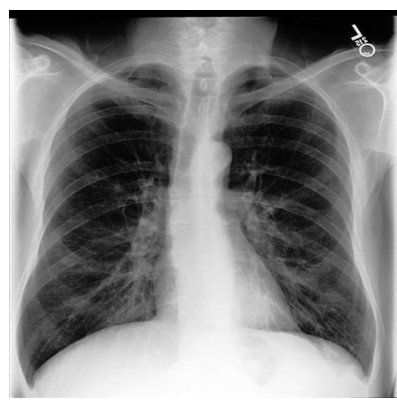

**A: Ground-Truth Report**

No focal consolidation, pleural effusion, or pneumothorax is seen. Heart and mediastinal contours are within normal limits. Lungs are again noted to be hyperinflated.

**B: SAE-Rad Report**

The chest radiograph demonstrates hyperinflated lungs with flattened hemidiaphragms and an enlarged retrosternal air space, consistent with chronic obstructive pulmonary disease. The lung fields appear clear without evidence of focal consolidation, pleural effusion, or pneumothorax. The cardiomediastinal silhouette is normal in size and contour. No acute osseous abnormalities are identified. Overall, there are no acute intrathoracic abnormalities noted on this examination. While small pulmonary nodules and subtle ground-glass opacities may be present, they are not well appreciated on this chest radiograph and would be better evaluated with CT imaging if clinically warranted.

☐ Shared findings between reports

☐ Finding missed by SAE-Rad

☐ Finding missed in ground-truth report

Figure 8: **Comparison between SAE-Rad Report and the Ground Truth (4)**. In this report, SAE-Rad captures nearly all elements from the ground truth. Both reports note hyper inflation in the lungs, no evidence of focal consolidation, pleural effusion, or pneuomothorax. Additionally, both reports note the normal size and shape of the cardiomediastinal silhoutte. SAE-Rad adds the presence of flattented hemiadiapharghms, an enlarged retrosternal air space, observes small pulmonary nodules, and the lack of acute osseous abnormalities. SAE-Rad misses the comparison to the prior patient history, where lungs were previously noted as hyperinflated.

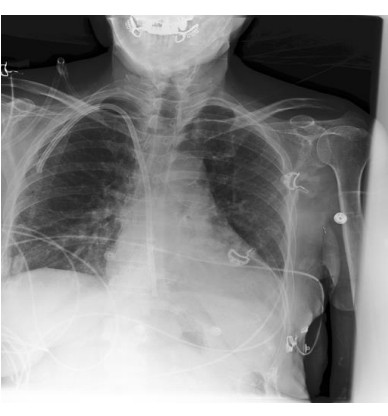

Shared findings between reports

Finding missed by SAE-Rad

Finding missed in ground-truth report

**A: Ground-Truth Report**

Continued mild cardiomegaly. Possible sml pleural effusions. Vasc congestion and mild interstitial edema. No focal consolidation. As compared to the prior study, there is interval minimal increase in vascular congestion. Cardiomediastinal silhouette is stable. There is no change in the appearance of the dialysis catheter. Small bilateral effusions are most likely present. There is no pneumothorax. No new consolidations to suggest infectious process demonstrated.

**B: SAE-Rad Report**

A central venous catheter is identified with its tip appropriately positioned in the superior vena cava or at the cavoatrial junction. There are small bilateral pleural effusions noted, with no evidence of pneumothorax. The lung fields demonstrate increased pulmonary vascular markings and interstitial thickening, consistent with mild to moderate pulmonary vascular congestion and edema. The cardiac silhouette appears enlarged, which is consistent with the findings from previous examinations and may suggest a waxing and waning pericardial effusion. No consolidation or focal airspace opacities are evident. The mediastinal contours are unremarkable, and there are no displaced rib fractures visible.

Figure 9: **Comparison between SAE-Rad Report and the Ground Truth (5)**. The presence of a catheter, as well as small bilateral pleural effusions, an enlarged cardiac silhoutte, and the presences of medium pulmonary vascular congestion with edema is shared across both reports. Both reports also note a lack of pneuomothorax, focal consolidations, and focal airspace opacities. SAE-Rad adds more details on the positioning of the catheter, as well as commenting on the normal shape of the mediastinal contours with no apparent displaced rib fractures. SAE-Rad misses the comparative statements with respect to prior studies in relation to the volume of vascular congestion, the changes in the cardiomediastinal silhouette, and the lack of change of position of the dialysis catheter between studies.

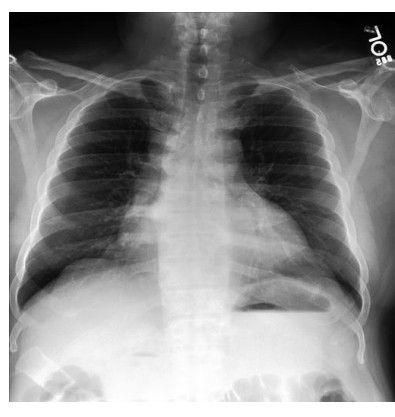

## A: Ground-Truth Report

In comparison with the study of ___, there is little change and no evidence of acute cardiopulmonary disease. No pneumonia, vascular congestion, or pleural effusion. The cardiac silhouette is at the upper limits of normal in size or slightly enlarged.

## B: SAE-Rad Report

The cardiac silhouette and mediastinal contours appear prominent, consistent with previous examinations. The heart size is at the upper limits of normal to mildly enlarged, without evidence of severe cardiomegaly. Lung fields are clear and well-expanded, showing no focal consolidations, effusions, or pneumothorax. There are no signs of acute pulmonary pathology or fluid overload. The absence of pleural effusions and pulmonary edema suggests that the cardiac enlargement is not currently associated with significant congestive heart failure. Overall, these findings represent a stable appearance compared to prior studies, with no acute cardiopulmonary process identified.

Shared findings between reports

Finding missed by SAE-Rad

Finding missed in ground-truth report

Figure 10: **Comparison between SAE-Rad Report and the Ground Truth (6)**. In this example, SAE-Rad captures a significant portion of features described in the ground truth report. Both reports note the normal size of the heart, the lack of pleural effusions or acute cardiopulmonary processes, and the stability of appearance with respect to previous exams. SAE-Rad correctly describes the clear appearance of the lungs, as well as the prominence of the mediastinal contours and cardiac silhoutte. Additionally, the lack of severe cardiomegaly, pneumothorax or focal consolidations is noted by SAE-Rad. Comparatively, SAE-Rad fails to note the lack of pneumonia and vascular congestion.

## A: Ground-Truth Report

The patient is status post median sternotomy as well as pacemaker placement with leads terminating in right atrium and ventricle. There is also a aortic valve prosthesis. The heart size remains normal. There are no focal opacities concerning for an infectious process. No pleural effusion and no pneumothorax.

## B: SAE-Rad Report

The chest radiograph demonstrates clear lung fields without evidence of focal consolidations, pleural effusions, or pneumothorax, suggesting the absence of acute pneumonia or other significant pulmonary pathology. The cardiac and mediastinal contours appear normal, with no signs of cardiomegaly or mediastinal widening. A permanent pacemaker is noted with leads appropriately positioned in the right atrium and right ventricle, unchanged from previous studies. The metallic component of an aortic valve prosthesis and sternotomy wires are again visualized, consistent with the patient's surgical history. Overall, the cardiopulmonary structures are stable compared to prior examinations, with no acute cardiopulmonary abnormalities identified.

- Shared findings between reports
- Finding missed by SAE-Rad
- Finding missed in ground-truth report

Figure 11: **Comparison between SAE-Rad Report and the Ground Truth (7)**. SAE-Rad correctly identifies all features in the ground truth. The lack of focal consolidations, pleural effusions and pneumothorax is noted in both reports. Additionally, SAE-Rad describes the normality of the cardiac and mediastinal contours, as well as the stability of the cardiac structures. SAE-Rad correctly identifies reported instrumentation, noting the pacemaker, sternotomy wires in line with the patient's history, as well as the location and presence of an aortic valve prosthesis. SAE-Rad additionally describes the lack of cardiomegaly or mediastinal widening, the stability of the pulmonary system, and the stability of the image with the respect to previous exams.

## C.2 IMAGE EXAMPLES OF MONOSEMANTIC FEATURES

In this section we showcase highest activating images for a number of features, as well as the corresponding feature explanations. We highlight the variety of features captured by SAE-Rad, from instrumentation features in Fig. 12, Fig. 13, Fig. 14, as well as visual features such as radiograph inversion in Fig. 16, pathology-related features in Fig. 17, and small details such as piercings in Fig. 18.

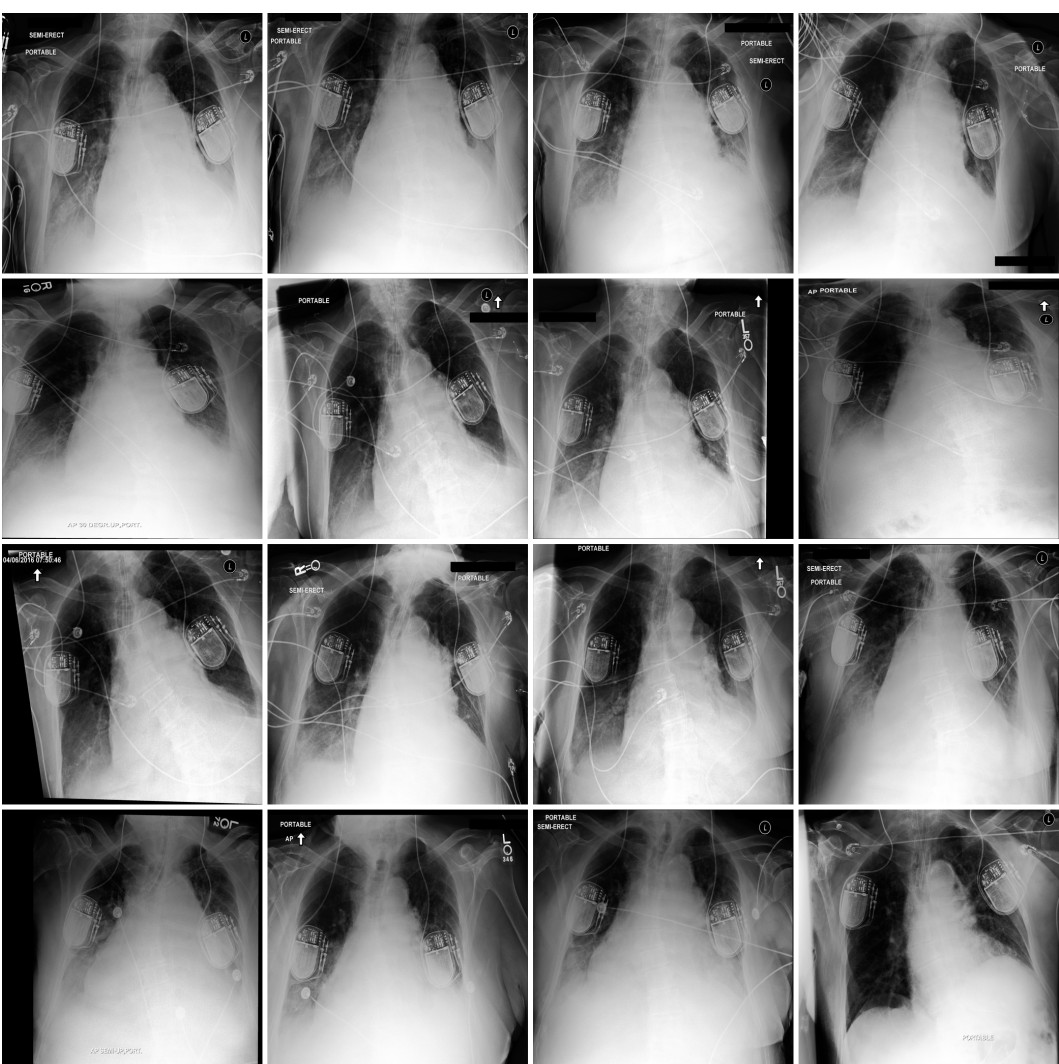

Figure 12: **Maximally activating images for a feature corresponding to deep brain stimulators which are typically used to treat Parkinson's disease.**

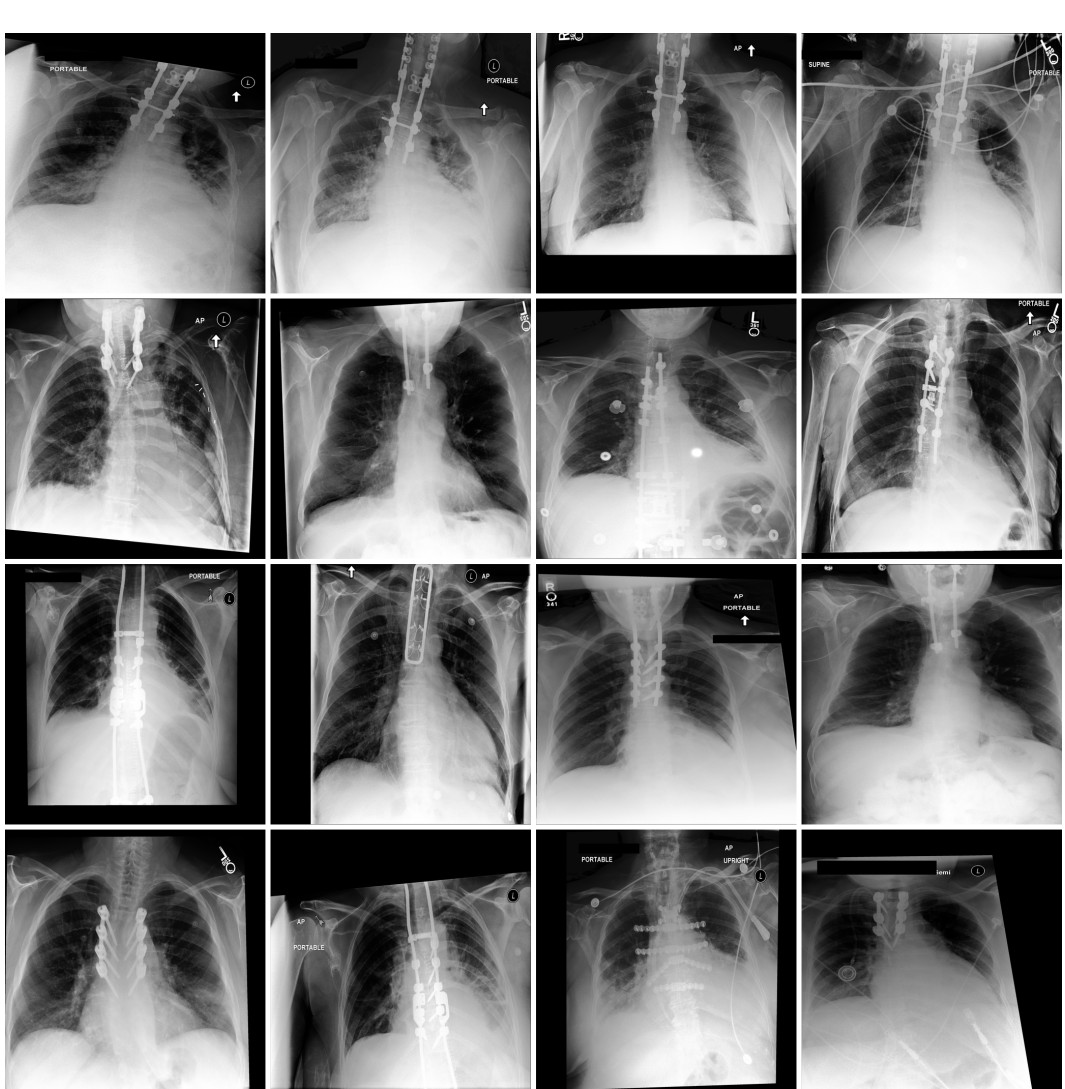

Figure 13: **Maximally activating images for a feature corresponding to orthopaedic rods and screws.**

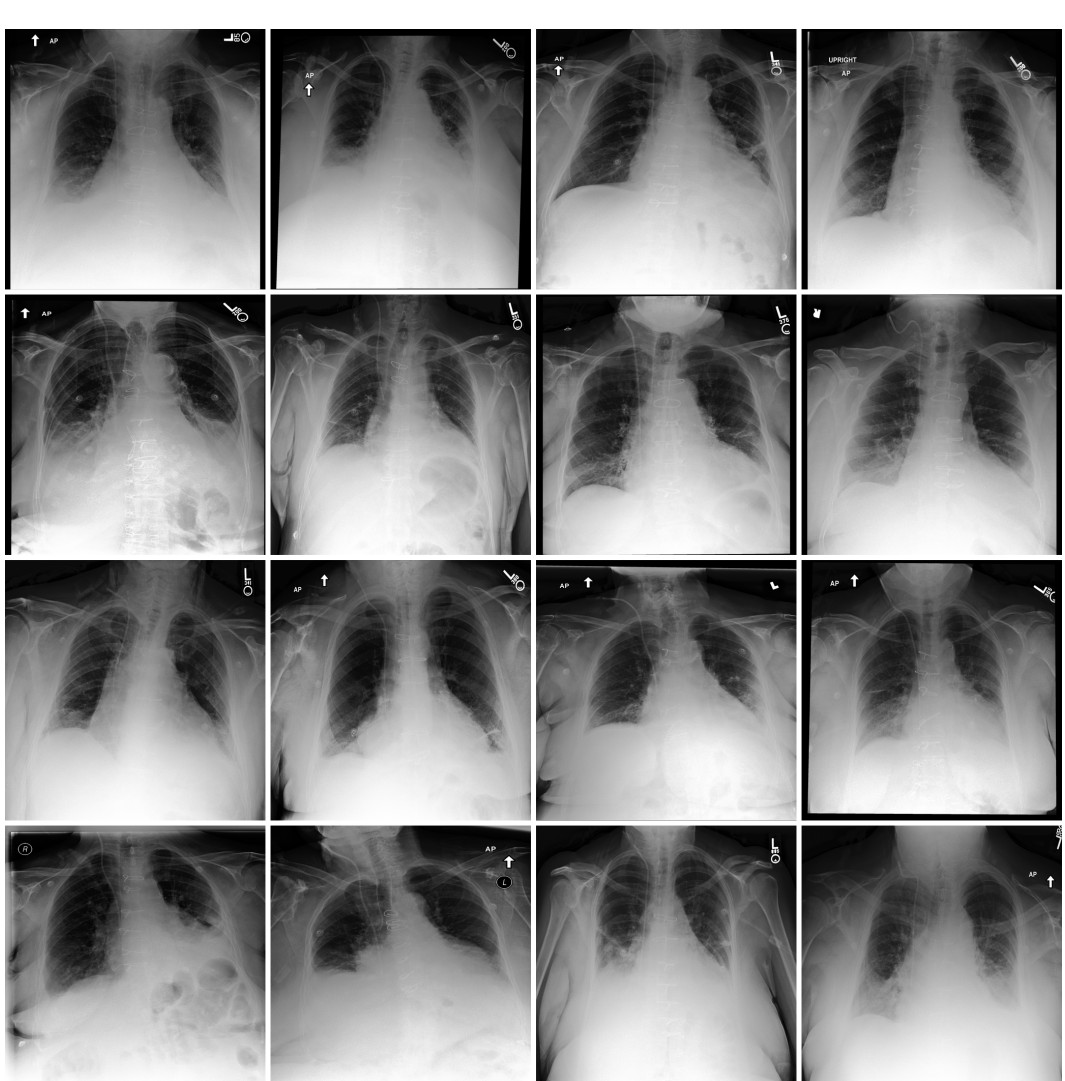

Figure 14: **Maximally activating images for a feature corresponding to right sided intra-jugular lines in the presence of sternotomy wires.**

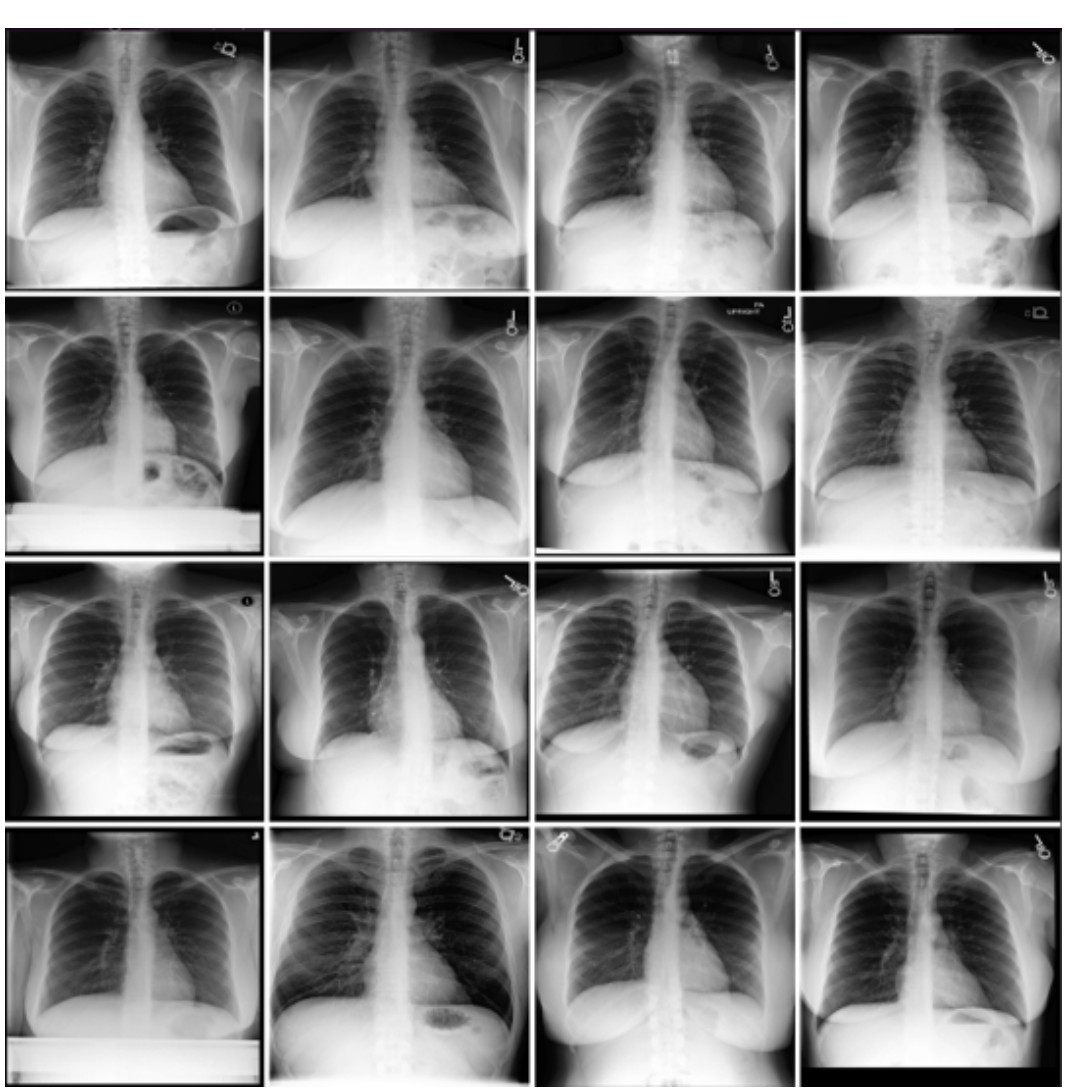

Figure 15: **Maximally activating images for a feature corresponding to female radiographs with no pathology detected.**

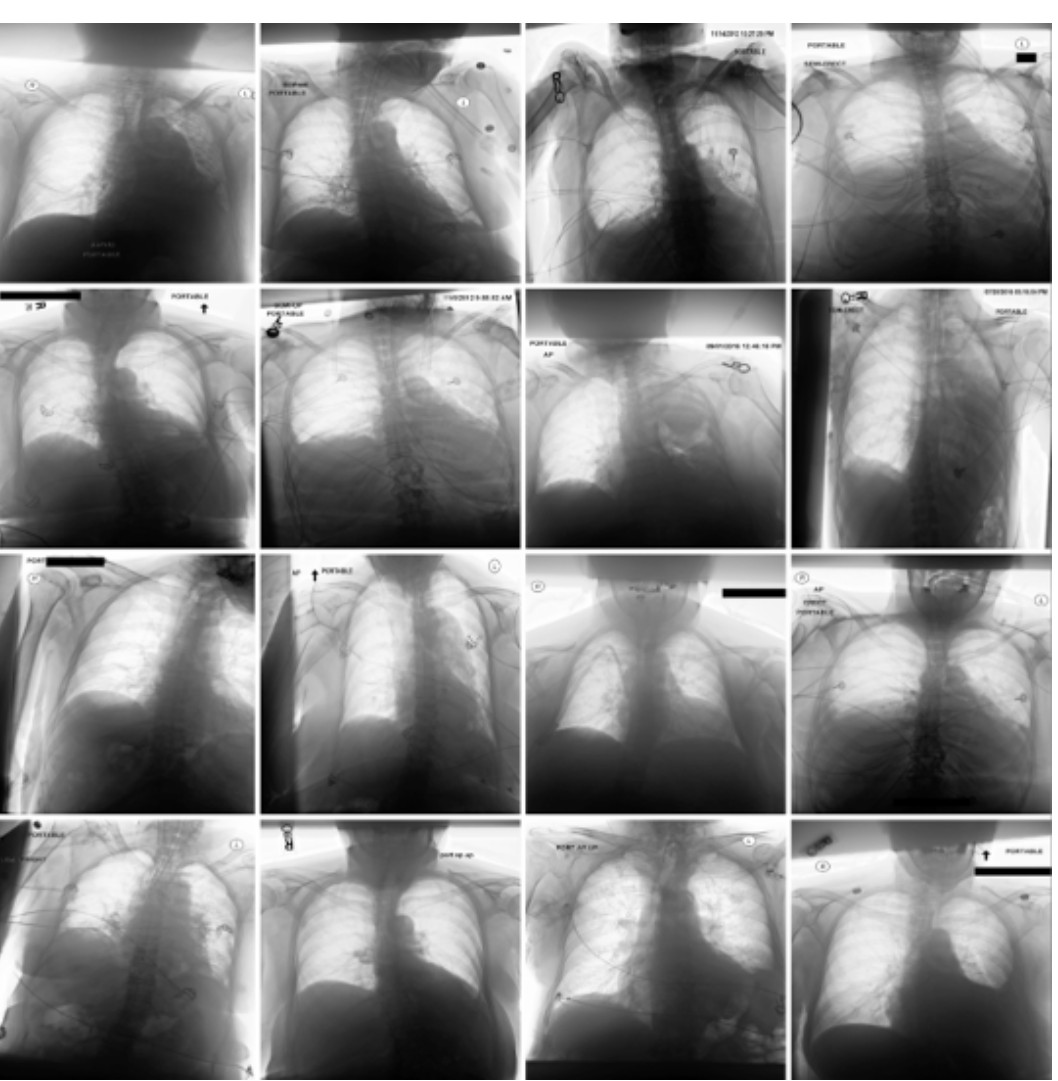

Figure 16: **Maximally activating images for a feature corresponding to inverted radiographs.**

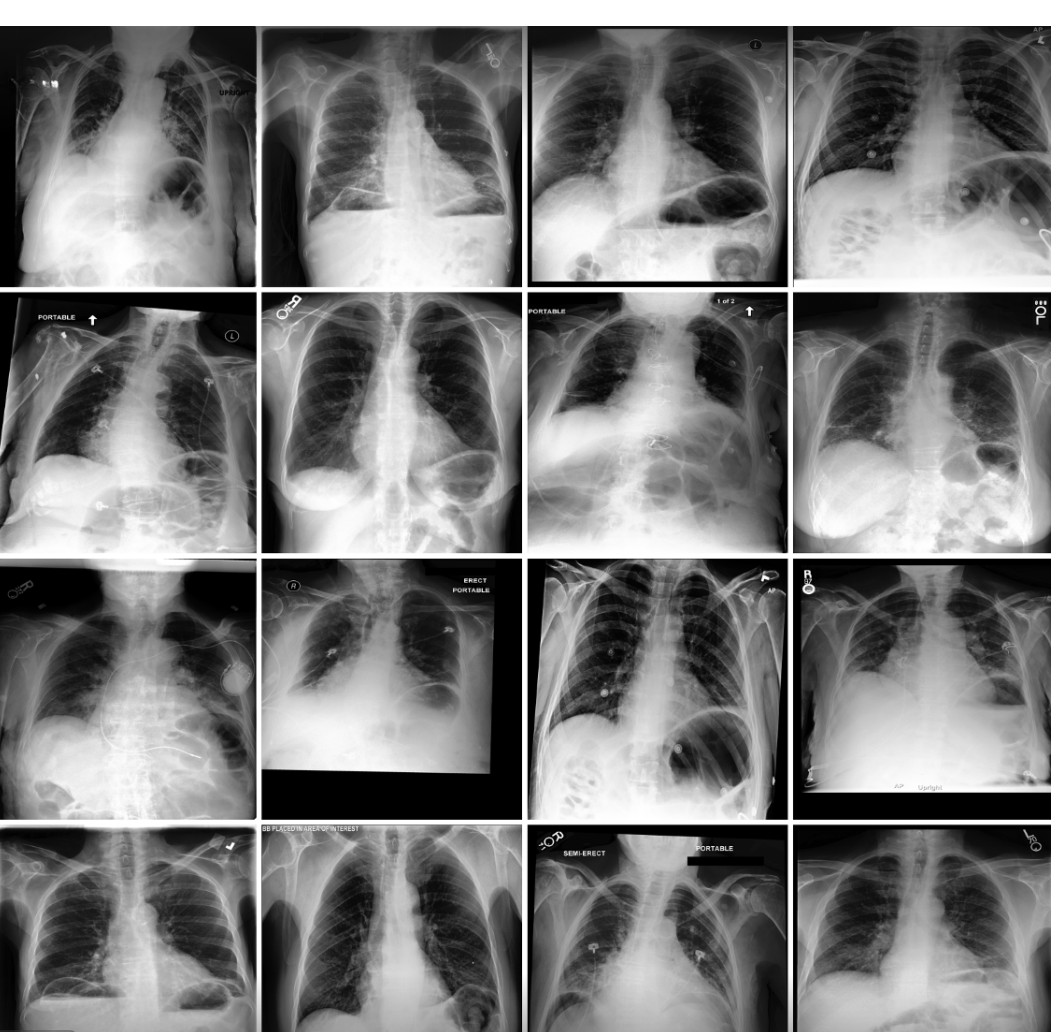

Figure 17: **Maximally activating images for a feature corresponding to bowel obstruction.**

1836
1837
1838
1839
1840
1841
1842
1843
1844
1845
1846
1847
1848
1849
1850
1851
1852
1853
1854
1855
1856
1857
1858
1859
1860
1861
1862
1863
1864
1865
1866
1867
1868
1869
1870
1871
1872
1873
1874
1875
1876
1877
1878
1879
1880
1881
1882
1883
1884
1885
1886
1887
1888
1889

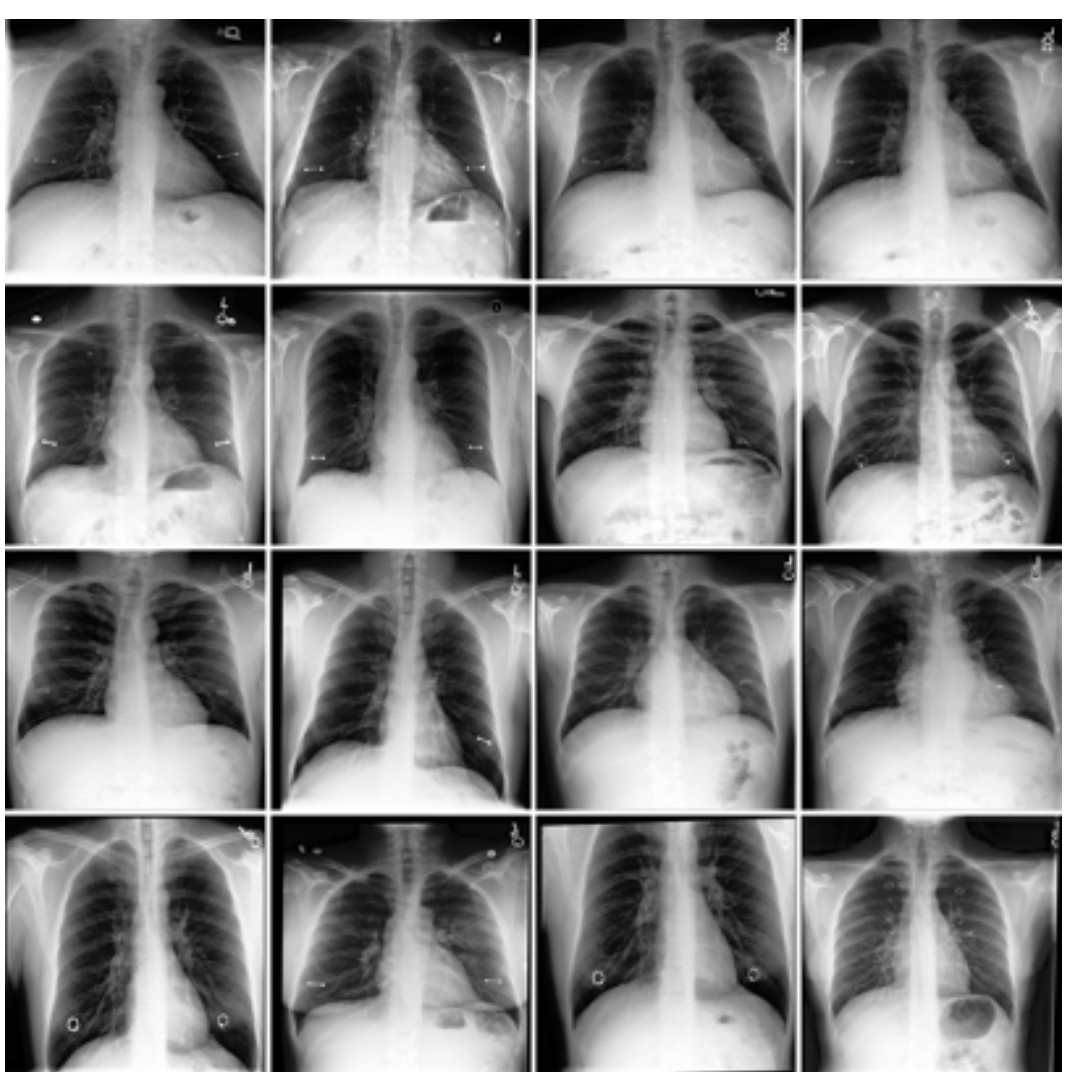

Figure 18: **Maximally activating images for a feature corresponding to piercings.**

## D Image-based feature localization implementation

**Model Training**    In order to visualise intervention effects, we trained a conditional denoising diffusion probabilistic model (DDPM) Ho et al. (2020) to generate chest radiographs given Rad-DINO tokens. Our DDPM architecture is a three block, 2D UNet, where each block consists of two residual sub-units with SiLU activation and group normalisation. The downward path consists of a first block of 2D convolutions, followed by spatial self-attention for the second and third blocks. The upward path consists of two spatial self-attention blocks followed by upsampling for the last block. The DDPM is conditioned by linearly projecting the CLS token $\mathbf{z} \in \mathbb{R}^{768}$ to a vector $\mathbf{v} \in \mathbb{R}^{1024}$, broadcasting the result to a 2D tensor $\mathbf{j} \in \mathbb{R}^{128 \times 128}$, and concatenating $\mathbf{j}$ channel-wise with a sample $\mathbf{n} \in \mathbb{R}^{128 \times 128}$ as an input to the UNet. The DDPM was trained for 330 epochs using a 90/10 train/val split of the MIMIC-CXR dataset, with a batch size of 256 and an AdamW optimizer with a learning rate of 0.001. We follow standard diffusion model training procedures by randomly sampling timesteps $\{t \in \mathbb{Z} | 0 \leq t \leq 1000\}$, noising an image $\mathbf{x}_0 \subset \mathbf{X}$ for $t$ steps to obtain noised image $\mathbf{x}_t = \sqrt{\alpha_t}\mathbf{x}_0 + \sqrt{1 - \alpha_t}\epsilon$, where $\alpha_t$ is a scalar controlling the variance of $\mathbf{x}_t$ as a function of $t$, and $\epsilon \sim \mathcal{N}(\mathbf{0}, \mathbf{I})$. The objective function can be defined over the mean squared error between diffusion model $\epsilon_\theta$ outputs conditioned on class tokens $\mathbf{z}$:

$$\mathcal{L} = \arg\min_\theta \mathbb{E}_{\mathbf{x}_0 \sim q(\mathbf{x}_0), t \sim U(0,1000), \epsilon \sim \mathcal{N}(0,\mathbf{I})} \left[ \|\epsilon_\theta^{(t)}(\sqrt{\alpha_t}\mathbf{x}_0 + \sqrt{1-\alpha_t}, \mathbf{z}) - \epsilon\|_2^2 \right] \tag{12}$$

Training was implemented using PyTorch (Paszke et al., 2019) and the `diffusers` (Huggingface, 2023) package on a single LambdaLabs 8xNVIDIA H100 cluster.

**Intervention Experiments**    We define a qualitative experiment to assess the visual interpretability of SAE-Rad features by evaluating whether activating or deactivating particular SAE features results in the expected visual changes. These inteventions can be defined using a constant reassignment operator

$$\mathsf{do}(i, \beta\,; \mathbf{h}(\mathbf{x})) : \mathbf{h}(\mathbf{x}) \mapsto \tilde{\mathbf{h}}(\mathbf{x})_{i,\beta} \tag{13}$$

which sets a particular feature in the SAE activation $\mathbf{h}(\mathbf{x})$ at index $i$ to a desired constant value $\beta$. Following Eq. (9), a given image token $\mathbf{z}$ may be reconstructed into an intervention token $\tilde{\mathbf{z}}_{i,\beta}$ by decoding it's intervened-upon activation $\tilde{\mathbf{h}}(\mathbf{x})_{i,\beta}$

$$\tilde{\mathbf{z}}_{i,\beta}(\mathbf{h}(\mathbf{z})) := W^{\mathrm{dec}}\mathsf{do}(i, \beta; \mathbf{h}(\mathbf{z})) + \mathbf{b}^{\mathrm{dec}}. \tag{14}$$

Given a feature of interest $i$, intervention quantity $\beta$, and token $\mathbf{z}$, we can compute a single "counterfactual" image $\tilde{I}_{i,\beta}$ by reverse denoising $\tilde{\mathbf{z}}_{i,\beta} + \delta$ using $\hat{\epsilon}_\theta$, where $\delta = \hat{\mathbf{z}} - \mathbf{z}$ accounts for the reconstruction error inherent to the SAE between the original token $\mathbf{z}$ and reconstructed $\hat{\mathbf{z}}$.

Using this framework, we assessed the counterfactual cyclic consistency for a given feature by performing two sequential interventions: we generated $\tilde{I}_{i,\beta}$ by *activating* feature $i$ by $\beta$ for a token $\mathbf{z}$, and subsequently reverted the effects of the intervention by *deactivating* $\beta$ at $i$ in $\tilde{\mathbf{z}}_{i,\beta}$ to generate $\tilde{I}_{i,0}$. Intuitively, $|\tilde{I}_{i,0} - \tilde{I}_{i,\beta}|$ should represent the visual concepts of interest. In our experiments, we empirically set $\beta = 15$, and select features 311 (pacemaker feature) and 162 (cardiomegaly) our features of interest. Features were empirically identified by reviewing highest activating feature subsets for our final SAEs. Results are showcased in Fig. 4.

## E Reader study

In addition to qualitative analyses performed in Section 5.1, we performed a reader study to assess radiologist opinions of generated reports compared to currently reported methods. We followed the reader study design from MAIRA-2 Bannur et al. (2024) - in summary, we presented a radiologist with a radiograph, the indication for the exam, and the generated report, and subsequently tasked the radiologist with assessing reports sentence by sentence, where the radiologist could edit, delete or maintain the sentences. Where changes were made, we additionally tasked the radiologist to categorise errors as one or multiple of "omission", "misclassification", "overspecific", "incorrect location", and "other". We omitted the category "incorrect progression", as we do not include past radiographs for consideration as part of our reader assessment. Additionally, unlike the MAIRA-2

reader study, errors could be classified as multiple error categories, which allowed for increased granularity in labelling. Finally, errors were categorised according to their clinical implications - errors could be "significant", "minor", or "none". The custom user interface (UI) used for the study is showcased in Fig. 19. For full details on error taxonomy, we refer the reader to Bannur et al. (2024).

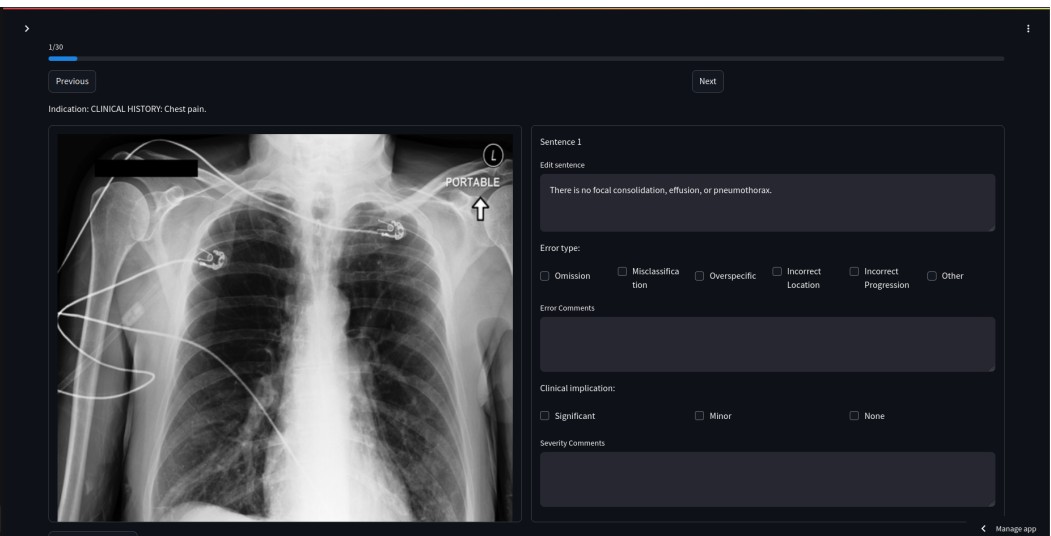

Figure 19: **Custom UI for radiograph evaluation**.

Overall, a radiologist (ST3) reviewed twenty unique radiographs with findings section extracted from the ground-truth closest report baseline described in Appendix B.6, as well as findings generated by CheXagent and SAE-Rad; in total, sixty individual reports were assessed. The radiologist was blinded to the model that findings originated from during the study.

### E.1 RESULTS

Overall, a total of 328 sentences were analysed, with an average of 5.5 sentences per report (CheXagent: 5.0, SAE-Rad: 5.9, Baseline: 5.5). Of 328 sentences, 240 (73%) sentences were edited or deleted, with 88 (27%) sentences maintained as is. Each report, on average, required 4.0 number of edits. Split by model, CheXagent had 74 (74%) sentences with edits, the baseline totalled 85 (77%) sentences with edits, and SAE-Rad totalled 81 (69%) sentences with edits.

Of sentences that required edits, 128 (68%) sentences overall had no clinical impact, 72 (30%) had a minor impact, and 30 (12%) had a significant impact. By model, CheXagent had 38 (51.4%) sentences with edits with no clinical impact, 23 (31.0%) sentences with minor impact and 13 (17.5%) sentences with significant clinical impact, the baseline had 44 (51.7%) sentences with edits with no clinical impact, 30 (35.3%) sentences with minor impact and 11 (13%) sentences with significant clinical impact, and SAE-Rad had 56 (69.1%) sentences with edits with no clinical impact, 19 (23.5%) sentences with minor impact and 6 (7.4%) sentences with significant clinical impact.

Sentences with edits were manually reviewed to assess their hallucinatory potential based off of the radiologist's feedback. We define hallucination as the presence of a clinical fact that is not verifiable or incorrect. This obviates edits which, for example, arise due to referencing previous studies, or edits resulting from minor grammatical differences or re-wording of clinical facts. Overall, we find that 124 (53.4%) of edits were classified as hallucinations, whilst 108 (46.5%) of edits were not classified as hallucinations. Of these edits, SAE-Rad produced the fewest hallucinations, with the baseline generating the most. Specifically, SAE-Rad produced 50 (64.1%) non-hallucinatory sentences, and 28 (35.9%) sentences that were classified as hallucinations. CheXagent produced 30 (41.1%) non-hallucinatory sentences, and 43 (58.9%) sentences that were classified as hallucinations. The baseline method produced 28 (34.5%) non-hallucinatory sentences, and 53 (65.4%) sentences that were classified as hallucinations.

In total, 160 (46.6%) errors were classified partly as "other", representing the majority class. 52 (15.1%) errors were omissions, 59 (17.2%) errors were mis-classifications, 57 (16.6%) errors were overspecific, and 15 (4.4%) errors were incorrect location. Error type and severity distribution are visualised in Fig. 20. Notably, whilst many "other" type errors were observed, the majority of these errors carried no clinical risk. In contrast, the "omission" and "misclassification" categories convey much higher risk, with a comparatively higher proportion of minor and significant severity categories. Overall, for sentences with edits, each sentence was labelled with an average $1.40 \pm 0.68$ error categories (SAE-Rad: $1.42 \pm 0.70$, CheXagent: $1.35 \pm 0.69$, baseline: $1.42 \pm 0.66$ error categories per edited sentence).

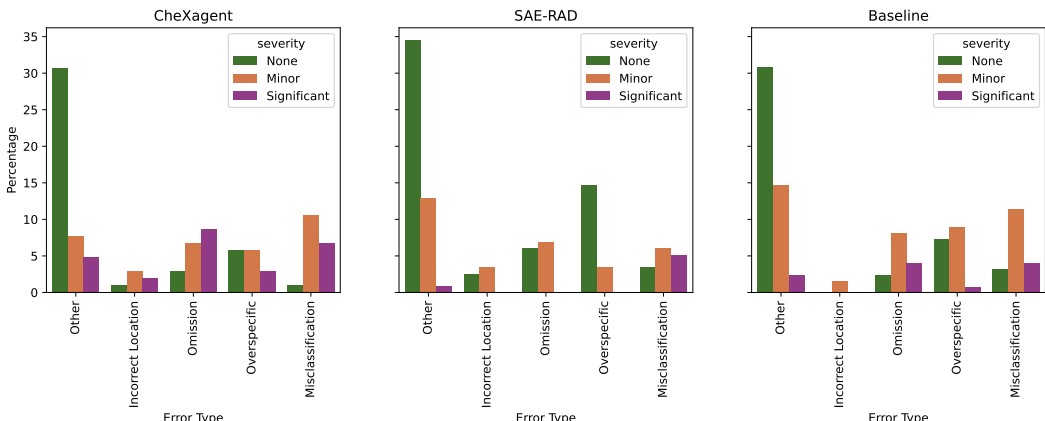

Figure 20: **Distribution of error types for different models**. Notably, SAE-Rad achieves the lowest number of "significant" errors in comparison to alternate methods.

Overall, SAE-Rad requires 7% fewer edits than the baseline, and 5% fewer edits than CheXagent. Additionally, significant clinical impact error rates are almost half for reports generated using our method compared to other models, whilst maintaining comparable minor and no impact error types. Similarly to results reported by Bannur et al. (2024), the majority of errors have no clinical impact. Overall, this showcases the potential our proposed method for radiology report generation in a real clinical scenario.

## F COMPUTE EFFICIENCY COMPARISONS

In this section we provide and discuss the compute estimates for training and inference of SAE-Rad in comparison to both MAIRA 2 and CheXagent. In Table 4 we provide quantitative estimates of the compute efficiency - we estimate that SAE-Rad uses approximately 1000x less train compute, 100x less parameters and 4x less data. We note that while in this paper we combine the SAE text features into a report using Claude, our method enables the generation of reports using a much smaller LLM, or alternately without an LLM — for example, by concatenating the SAE feature text explanations into a report using only regex. This reduces the inference FLOPs and cost to a negligible quantity. In Table 4, we have estimated the lower bound inference compute requirements for SAE-Rad using this assumption.

|  | SAE-Rad | cheXagent | MAIRA 2 |
|---|---|---|---|
| Number of unique training images | 239,931 | 1,100,000 | 510,848 |
| Active training parameters | 76M | 8B | 7B |
| Estimated training Tera-FLOPs | 60,000 | NA | 100,000,000 |
| Estimated training cost | $3.25 | NA | $4000 |
| Estimated inference Tera-FLOPs per report | 0.1 | 8 | 7 |
| Estimated inference cost per report | $0.00005 | $0.001 | $0.001 |

Table 4: Comparison of compute resources for different models.

## G LIMITS OF AUTOMATED QUALITY ASSESSMENT FOR RADIOLOGY REPORTS

As described in Appendix B.2, NLG metrics such as BLEU-4 (Papineni et al., 2002), ROUGE-L (Lin, 2004), and METEOR (Banerjee & Lavie, 2005) do not account for the clinical relevance of the sentences composing a radiology report. Radiology-specific metrics such as the RGER score (Delbrouck et al., 2022) and the CheXpert F1 score (Smit et al., 2020) were consequently developed, seeking to improve upon these limitations by using radiology-specialized models as backbones to compute clinically relevant commonalities between generated and reference reports. However, CheXpert and RGER rely on pre-specified findings classes and error types (Bannur et al., 2024), which limits their ability to assess the quality of generated reports when elements outside these specified categories are present. RadFact (Bannur et al., 2024) was recently proposed to address this limitation; by relying on the logical inference capabilities of large language models (LLMs), RadFact evaluates the correctness and completeness of generated reports outside of pre-specified classes.

RadFact introduces two key metrics: RadFact logical precision and RadFact logical recall, henceforth denoted as RadFact precision and recall. RadFact precision calculates the proportion of generated sentences that are entailed by the ground-truth report, assessing the truthfulness of the model's outputs by penalizing hallucinations. RadFact recall measures the proportion of ground-truth sentences that are entailed by the generated report, evaluating the completeness of the generated report by penalizing omissions.

However, RadFact has several limitations. Firstly, it relies on an LLM to extract discrete statements from reports, which may introduce inaccuracies. The LLM must also perform bi-directional entailment verification—a challenge for models like Llama3-70B-Instruct (RadFact's backbone) given the complex medical reasoning in the reports. Furthermore, LLMs face the reversal curse (Berglund et al., 2023): they excel at forward relationships (e.g., 'France's capital is Paris') but struggle with reverse ones ('Paris is the capital of what country?'), compounding the difficulty of bi-directional entailment verification. A particularly important limitation of RadFact is that it relies on the assumption that the ground-truth report used as a basis for evaluation is a gold-standard label which exhaustively captures clinically relevant detail.

Therefore, in cases where a predicted report correctly includes findings present in the image but omitted from the ground-truth report, RadFact precision will drop. This means that the metric may not fully reflect the model's ability to generate clinically accurate and comprehensive reports, when the ground-truth reports are incomplete.

We illustrate potential pitfalls of RadFact in Fig. 21 below. In this case, the ground-truth report fails to report several clinically relevant details observed in the SAE-Rad report, including, for example, the normal size of the cardiac silhouette, and the increased density in the right upper lung field. We provide additional examples of this phenomenon in Appendix C.1.

We note that the level of detail for radiographic reports is likely impacted by multiple factors, such as report indication, patient history, urgency of the request, and care setting of the radiograph. Given we perform evaluation on MIMIC-CXR —a dataset of radiographs captured at an emergency department (Johnson et al., 2019)— SAE-Rad is likely to report clinically correct and relevant statements which are not explicitly mentioned by radiologists. Therefore, reported metrics must be interpreted with care, and may be underestimates of actual clinical factuality and relevance of generated reports.

## H ABLATION EXPERIMENTAL SETUP

In this section we present the experimental setup for our ablation studies. We evaluated six SAEs on RadFact across three expansion factors and two sparsity levels. We evaluated the SAEs using a set of 60 randomly selected radiographs from the MIMIC-CXR test split. RadFact was evaluated using the "Llama3-70B-Instruct" model. Additionally, we report NLG and clinical metrics to assess the difference in quality of reports across models. In this study, we used an earlier prompting template to generate feature explanations and subsequent reports. Overall, we note that this change resulted in slightly worse quantitative metrics than those that were obtained using the final prompting strategy.

All SAEs were trained for 200,000 steps with a batch size of 2048. Models were trained using an Adam optimizer with no weight decay and a learning rate of $5 \times 10^{-5}$, as well as a linear warm-up

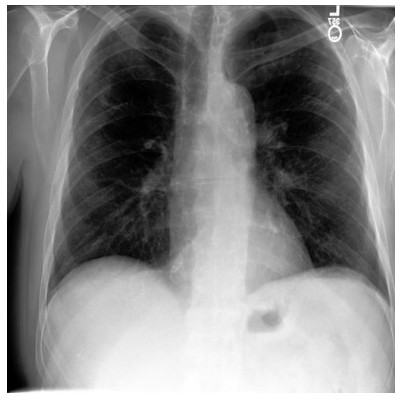

**A: Ground-Truth Report**

In comparison with study of ___, there are fibronodular changes again seen in the upper zones, consistent with the clinical diagnosis of sarcoidosis. No evidence of acute focal pneumonia, vascular congestion, or pleural effusion.

**B: SAE-Rad Report**

The chest radiograph demonstrates persistent bilateral upper lung reticular nodular opacities, consistent with the patient's known history of sarcoidosis. There are small, localized areas of increased density in the right upper lung field, which may represent calcifications or scarring from previous infection or healed processes. The lung fields are otherwise clear, without evidence of focal consolidations, pleural effusions, or pneumothorax. The cardiac silhouette is normal in size, and mediastinal contours appear unremarkable. No signs of pulmonary edema or acute cardiopulmonary process are observed. The overall appearance is stable compared to prior imaging, with no new or worsening focal or multifocal opacities to suggest acute infection or exacerbation of the underlying condition.

☐ Shared findings between reports

☐ Finding missed by SAE-Rad

☐ Finding missed in ground-truth report

Figure 21: **Example image for which an SAE-Rad generated report has significantly more detail than the ground-truth.** SAE-Rad captures nearly all features noted in the ground-truth such as: fibronodular changes, a diagnosis of sarcoidosis and pleural effusions. However SAE-Rad also notes the lack of pneuomothorax, pulmonary edema and the increased density in the right upper lung field. Despite the clinical relevance of these elements this report likely scored poorly on RadFact due to the incompleteness of the ground-truth.

of the learning rate for the first 1% of training, linear warm-down of learning rate for the last 20% of training, and an L1 coefficient warm-up for the first 5% of training.

We summarise experimental differences between runs in Table 5, as well as resultant L0 and explained variance metrics. The L0 and explained variance metrics in Table 5 provide insights into the quality of our SAEs. Notably, the "dense" variants exhibit higher L0 values, indicating reduced sparsity, but achieve greater explained variance. The different variants therefore represent different points on the Pareto frontier between sparsity and reconstruction quality.

Table 5: Varying hyperparameters and resulting metrics for the six SAEs.

| | Expansion factor | L1 coefficient | L0 ↓ | Explained variance (%) ↑ |
|---|---|---|---|---|
| ×128 | 128 | $6 \times 10^{-3}$ | 12.0 | 90.9 |
| ×64 | 64 | $8 \times 10^{-3}$ | 13.6 | 84.3 |
| ×32 | 32 | $9 \times 10^{-3}$ | 15.1 | 79.2 |
| ×128_dense | 128 | $3.75 \times 10^{-3}$ | 26.0 | 93.2 |
| ×64_dense | 64 | $5 \times 10^{-3}$ | 28.4 | 88.5 |
| ×32_dense | 32 | $5.63 \times 10^{-3}$ | 31.8 | 84.7 |

Results for clinical and NLG metrics for the ablation study are presented in Table 6. Overall, we find that the ×64 SAE (i.e SAE-Rad) is the maximally performant model for 6/9 of the metrics, with an additional 2/9 metrics being the second best of all models.

## I    INTERACTIVE EXPLORATION OF SAE FEATURES

To investigate the geometry of SAE features (Mendel, 2024), we embed the feature directions from the SAE decoder into a 2-dimensional space using a UMAP (McInnes et al., 2018) (for which we set the number of neighbors to 50 and the minimum distance to 0.05). This analysis allows us to

Table 6: **Report generation performance with clinical metrics for ablation study radiographs.** BL4 = BLEU-4, RG-L = ROUGE-L, MTR = Meteor. Ma-5 (Macro-F1-5), Ma-14 (Macro-F1-14), Mi-5 (Micro-F1-5), and Mi-14 (Micro-F1-14) represent the clinical CheXbert labeler scores. Bolding represents best performance on the development set, and italics represent second best model for a given metric.

| Model | RadFact ↑ | NLG Metrics ↑ | | | Clinical Metrics ↑ | | | | |
|---|---|---|---|---|---|---|---|---|---|
| | F1 | BL4 | RG-L | MTR | RGER | Ma-5 | Ma-14 | Mi-5 | Mi-14 |
| ×32 dense | 30.22 | 1.5 | *18.6* | 23.3 | *19.5* | 51.8 | 33.2 | 55.0 | 52.6 |
| ×64 dense | 29.74 | *1.9* | **19.5** | *23.9* | *19.5* | **57.8** | 31.9 | *58.7* | *55.5* |
| ×128 dense | 29.56 | 1.7 | 16.1 | 22.8 | 17.3 | 46.9 | **33.9** | 57.5 | 53.7 |
| ×32 | 29.46 | 2.2 | 16.2 | 23.7 | 16.6 | 44.2 | 25.3 | 53.5 | 48.8 |
| ×64 | **33.83** | **2.4** | 17.1 | **24.4** | **20.7** | *54.7* | *33.6* | **58.9** | **56.8** |
| ×128 | *32.18* | 1.8 | 16.4 | 24.1 | 18.9 | 45.2 | 25.7 | 55.8 | 49.8 |

visualize whether features which have semantically similar descriptions are clustered in the latent representations of the SAE. Text information is not used to train the Rad-Dino vision encoder, SAEs, or the UMAP projections of the features. Therefore, assessing the clustering patterns based on text descriptions allows us to independently validate the quality of learned representations. We develop interactive tooling to evaluate the results of this analysis. Clicking on each point will show the text description of the feature and load highest activating images. The tool can be accessed using the following link: `https://scatter-plot-app.vercel.app/`. For convenience, we also illustrate a number of well-clustered feature categories in Fig. 22.

It can be seen from Fig. 23 that the clustering enables the detection of possibly incorrect feature descriptions. The figure provides an example relating to a feature that captures the orientation of radiographs, rather than the currently described pathology. This approach can be used to ensure hallucination-free feature descriptions.

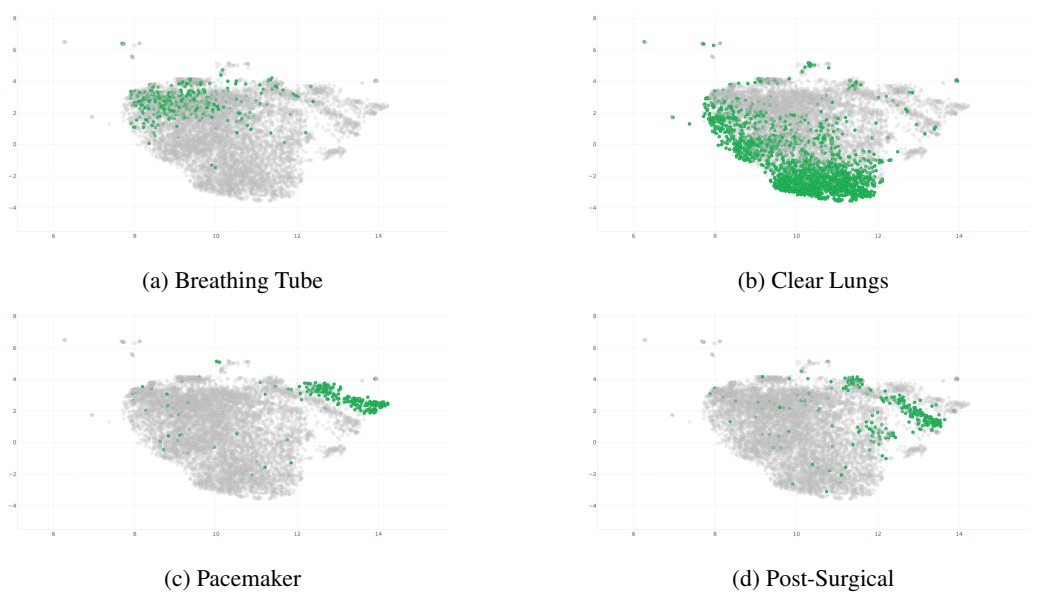

(a) Breathing Tube

(b) Clear Lungs

(c) Pacemaker

(d) Post-Surgical

Figure 22: **UMAP analysis for several feature categories.** Subfigures show examples of UMAP embeddings for (a) Breathing Tube, (b) Clear Lungs, (c) Pacemaker, and (d) Post-Surgical cases.

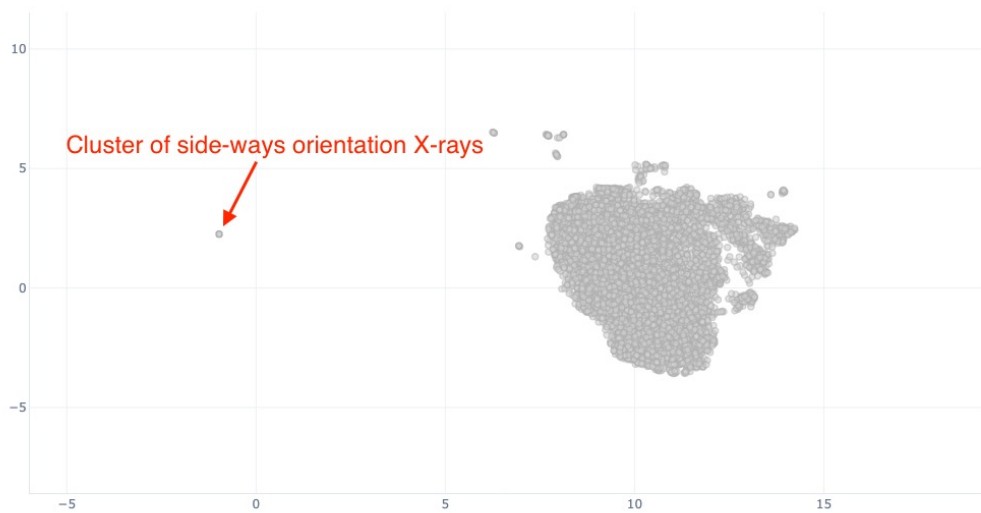

(a) Location of a cluster of sideways orientation X-rays within the UMAP.

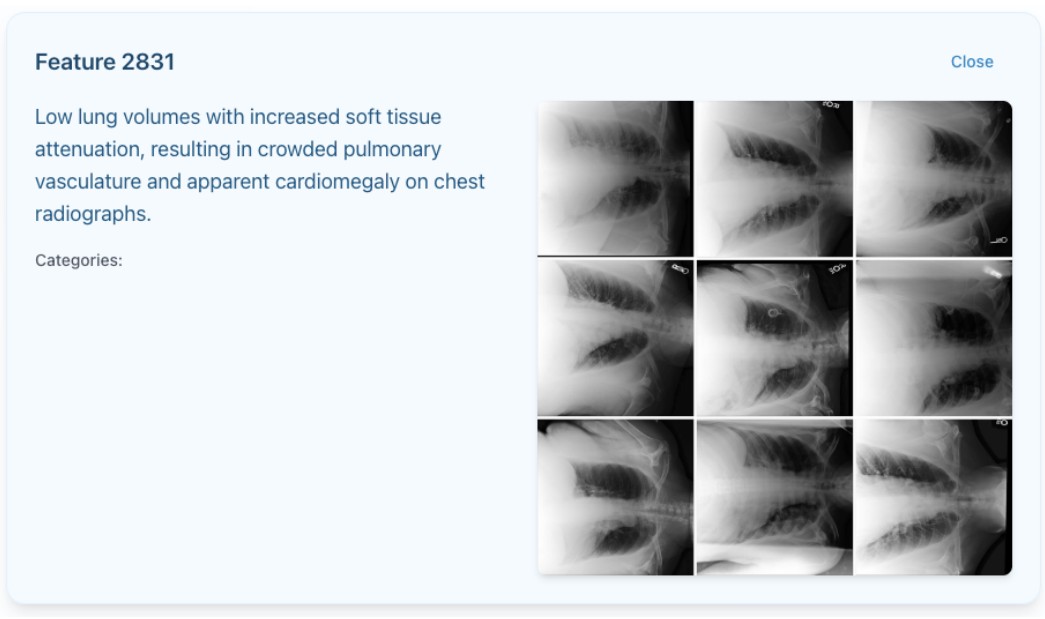

(b) An example feature from this cluster illustrating highest activating images with sideways orientation.

Figure 23: **Illustration of an orientation cluster.** Here we show an example of how an automated description can be manually visualized and corrected. In this the case the feature actually represents sideways orientation radiographs. To explore this cluster please visit the interactive tool here: `https://scatter-plot-app.vercel.app/`

## J  ADDITIONAL SAE-RAD COMPARISONS

We present additional comparisons to our method by collating reported clinical efficacy metrics and NLG metrics in Table 7. We find that SAE-Rad broadly outperforms existing methods across all clinical metrics.

Table 7: **Report generation performance on the official MIMIC-CXR test split on additional baselines.** BL4 = BLEU-4, RG-L = ROUGE-L. Ma-5 (Macro-F1-5), Ma-14 (Macro-F1-14), Mi-5 (Micro-F1-5), and Mi-14 (Micro-F1-14) represent the clinical CheXbert labeler scores as reported in the original studies. Bolding represents best performance in the current study or between the upper bound models.

| Model | NLG Metrics ↑ | | Clinical Metrics ↑ | | | |
|---|---|---|---|---|---|---|
| | BL4 | RG-L | Ma-5 | Ma-14 | Mi-5 | Mi-14 |
| GPT-4V Wu et al. (2023) | 1.9 | 13.2 | 19.6 | 20.4 | 25.8 | 35.5 |
| LLaVa-Med Wu et al. (2023) | 1.0 | 13.3 | 16.6 | 15.5 | 22.0 | 27.2 |
| CvT2Dist Codella et al. (2024) | 12.7 | 28.6 | - | 30.7 | - | 44.2 |
| LLaVa Codella et al. (2024) | 1.3 | 13.8 | 17.5 | 15.4 | 23.4 | 22.9 |
| GPT-4o finetune Codella et al. (2024) | 17.8 | 32.1 | 43.8 | 33.0 | 52.7 | 48.9 |
| GPT-4o mini finetune Codella et al. (2024) | 16.2 | **32.2** | 42.0 | 30.8 | 51.8 | 47.6 |
| R2GenGPT Wang et al. (2023b) | 13.4 | 16.0 | - | - | - | 38.9 |
| METransformer Wang et al. (2023a) | 12.4 | 29.1 | - | - | - | 31.1 |
| R2Gen Wang et al. (2023a) | 10.3 | 27.7 | - | - | - | 27.6 |
| R2GenCMN Wang et al. (2023a) | 17.0 | 19.1 | - | - | - | 27.8 |
| MSAT Yang et al. (2023) | 11.1 | - | - | - | - | 33.9 |
| KIUT Huang et al. (2023) | 11.3 | 28.5 | - | - | - | 32.1 |
| RGRG Tanida et al. (2023) | 12.6 | 26.4 | - | - | 54.7 | 44.7 |
| Flamingo-CXR Tanno et al. (2024b) | **29.7** | 10.1 | - | | **58.0** | 51.9 |
| SAE-Rad (x64) | 1.9 | 17.1 | **47.2** | **34.3** | 54.4 | **53.2** |

## K  EQUIVALENCE OF LOSS FUNCTIONS BETWEEN SAE ARCHITECTURES

Here, we show that the original SAE loss function in Eq. 3 is isomorphic to the loss introduced by (Conerly et al., 2024) (displayed in Eq. 7). We show that they are related by a fundamental group of symmetries in the design of SAEs. In particular, we construct a group of symmetries that preserve the reconstructed activations of the SAE and leave the (Conerly et al., 2024) loss invariant. The authors modified the sparsity-inducing part of the loss, which is specified below:

$$\mathcal{L}_{\text{sparsity}} = \sum_i \mathbf{h}_i(\mathbf{x}) \cdot \|W_{\cdot,i}^{\text{dec}}\|_2$$

where $W_{\cdot,i}^{\text{dec}}$ is the decoder weight matrix and $\mathbf{h}_i(\mathbf{x})$ is the activation value of the $i^{\text{th}}$ SAE feature.

For simplicity, we consider SAEs without any biases (though it should be noted that the explanation provided generalises to SAEs with biases). With this in mind, an SAE has the following architecture:

$$\mathbf{h}(\mathbf{x}_{\text{in}}) = \text{ReLU}(W^{\text{enc}}\mathbf{x}_{\text{in}})$$

$$\mathbf{x}_{\text{out}} = W^{\text{dec}}\mathbf{h}(\mathbf{x}_{\text{in}})$$

where $\mathbf{x}_{\text{in}} \in \mathbb{R}^n$ is the input activation vector, $W^{\text{enc}} \in \mathbb{R}^{m \times n}$ is the encoder matrix, $W^{\text{dec}} \in \mathbb{R}^{n \times m}$ is the decoder matrix, $\mathbf{h}(\mathbf{x}_{\text{in}})$ is the hidden activation of the SAE, and $\mathbf{x}_{\text{out}} \in \mathbb{R}^n$ is the reconstructed activation. Note that there are no normalisation constraints on the decoder matrix.

### K.1  MOTIVATION

Let us begin by highlighting the fact that multiplication by positive constants commutes with the ReLU activation function:

$$\text{ReLU}(\lambda x) = \lambda \text{ReLU}(x), \forall x \in \mathbb{R}, \lambda > 0$$

Motivated by this expression, we can define a second SAE with weights given by $\tilde{W}^{\text{enc}}, \tilde{W}^{\text{dec}}$ as follows:

$$\tilde{W}^{\text{enc}} = \text{diag}(\boldsymbol{\lambda})W^{\text{enc}}$$

$$\tilde{W}^{\text{dec}} = W^{\text{dec}}\text{diag}(\frac{1}{\boldsymbol{\lambda}})$$

where $\boldsymbol{\lambda} \in \mathbb{R}^m$ is now a vector. This SAE is identical to the original SAE when viewed as a function $\mathbf{x}_{\text{in}} \to \mathbf{x}_{\text{out}}$. The only difference is that the hidden activations $\tilde{\mathbf{h}}(\mathbf{x}_{\text{in}}) = \text{diag}(\boldsymbol{\lambda})\mathbf{h}(\mathbf{x}_{\text{in}})$ have been scaled by $\text{diag}(\boldsymbol{\lambda})$. Since this transformation does not change the output $\mathbf{x}_{\text{out}}$, it will not change the MSE reconstruction loss. This transformation will however change the $l_1$ sparsity loss. The $l_1$ loss changes by:

$$\tilde{l_1} = |\tilde{\mathbf{h}}(\mathbf{x}_{\text{in}})|_1 = |\text{diag}(\boldsymbol{\lambda})\mathbf{h}(\mathbf{x}_{\text{in}})|_1$$

It follows that if we do not impose the constraint of normalising the decoder weights, the SAE will use this symmetry in the limit $\boldsymbol{\lambda} \to 0$ to reduce $l_1$ to 0. This explains the necessity of normalising the decoder weights.

### K.2 EXPLANATION OF THE CONERLY ET AL. LOSS

Are there alternatives to normalising the decoder bias? What if instead we find a way to penalise the SAE for the transformation $\boldsymbol{\lambda} \to 0$ instead? We know that as $\boldsymbol{\lambda} \to 0$, $||W_{\cdot,i}^{\text{dec}}||_2 \to \infty$. Could we penalise the network by weighting the terms in the sum for $l_1$ by the corresponding term $||W_{\cdot,i}^{\text{dec}}||_2$? We know that the $l_1$ loss is given by:

$$l_1 = \sum_i |\mathbf{h}_i(\mathbf{x}_{\text{in}})|$$

We also know that the components $\mathbf{h}_i(\mathbf{x}_{\text{in}})$ transform in the opposite (covariant) way to decoder vectors $W_{\cdot,i}^{\text{dec}}$. In particular, the product $\mathbf{h}_i(\mathbf{x}_{\text{in}})||W_{\cdot,i}^{\text{dec}}||_2$ remains invariant under our transformation. We therefore define an invariant loss function as follows:

$$\mathcal{L}_{\text{sparsity}} = \sum_i \mathbf{h}_i(\mathbf{x}_{\text{in}}) \cdot ||W_{\cdot,i}^{\text{dec}}||_2$$

We have therefore penalised the SAE in such a way that there is no insentive to transform by any value of $\boldsymbol{\lambda}$; both the MSE loss and sparsity loss are now completely invariant. This is precisely the loss function introduced by Conerly et al. (2024). If we use this loss function, we have found a group of symmetries that preserve the MSE and sparsity losses. This group is isomorphic to the direct product of the group $(\mathbb{R}^+, \times)$ with itself $m$-times.

### K.3 EQUIVALENCE TO THE ORIGINAL LOSS

We now construct an equivalence to the old $l_1$ sparsity loss function. Suppose we have an SAE trained with the loss in Eq. 7. We are now at will to transform with whatever value of $\boldsymbol{\lambda}$ we like since they define a group of symmetries preserving the new loss. Let us pick $\boldsymbol{\lambda}$ as follows:

$$\boldsymbol{\lambda}_i = ||W_{\cdot,i}^{\text{dec}}||_2$$

By using this transformation, we define an equivalent SAE in which the decoder weights are now normalised. In addition the resulting (invariant) sparsity loss function has the form of the original loss function, the $l_1$ loss, since the decoder weights are now unit normalised.

This analysis shows that the Conerly et al. (2024) loss function is equivalent to the original SAE loss function but now has a natural invariance preserved under our symmetry group.

