# OpenReview forum: "An X-Ray Is Worth 15 Features: Sparse Autoencoders for Interpretable Radiology Report Generation"
_ICLR.cc/2025/Conference — Submitted to ICLR 2025_

### Official Review · Reviewer_hUc2 · 2024-10-28

**Soundness:** 3
**Presentation:** 2
**Contribution:** 3
**Rating:** 5
**Confidence:** 5

**Summary:**

This paper introduces SAE-Rad, a framework for generating radiology reports using Sparse Autoencoders (SAEs) to decompose image representations into human-interpretable features. SAE-Rad uses a hybrid gated SAE architecture to balance reconstruction accuracy and sparsity, extracting meaningful medical features. The extracted features are transformed into detailed reports by a pre-trained large language model (LLM) without requiring further fine-tuning. Experiments on the MIMIC-CXR dataset show that SAE-Rad achieves competitive performance on radiology-specific metrics, outperforming models like CheXagent with fewer computational resources.

**Strengths:**

1.Experimental Section: (1) The qualitative analysis is comprehensive, effectively demonstrating the capabilities of our model. (2) The ablation study is thoroughly designed.

2.The methodology is straightforward and effective, with a clear motivation. This approach directly decomposes image class tokens from a pre-trained radiology image encoder into human-interpretable features, which are then compiled into comprehensive radiology reports.

**Weaknesses:**

1.The novelty of the method is not thoroughly demonstrated. While applying Sparse Autoencoders (SAE) in vision-language models (VLMs) represents an innovative effort, the use of SAEs to enhance the interpretability of large language models (LLMs) is already a well-established area. Examples include OpenAI's "Scaling and Evaluating Sparse Autoencoders", which extends SAEs to GPT-4, and Google DeepMind's "Gemma Scope: Open Sparse Autoencoders Everywhere All At Once on Gemma 2".
2.The comparison with existing methods is limited. Table 1 only compares three methods: CheXagent, MAIRA-1, and MAIRA-2, lacking a broader comparison with other open-source methods that generate reports using large language models (LLMs), such as R2GenGPT, RaDialog, and XrayGPT. Although these methods may not utilize the Clinical Metrics evaluation indicators employed in this work, they are open-source, and some provide pre-trained model weights. Why not attempt to replicate these indicators on these open-source methods?
3.The experimental results lack persuasiveness. Although the paper notes that NLG Metrics are not sufficiently comprehensive for evaluating radiological reports, the vast majority of report generation studies still rely on NLG Metrics as the primary basis for evaluation. However, the scores achieved by this work on NLG Metrics are not competitive; with a BLEU-4 score of 1.9, it falls significantly short when compared to other LLM-based radiological report generation efforts, such as RaDialog with a 9.5 and R2GenGPT with a 13.4.
4.The experimental dataset is limited. The quantitative comparisons are conducted using only one public dataset, MIMIC-CXR. However, two other commonly used public datasets for medical report generation are IU-Xray and openi. The paper R2GenGPT, XrayGPT incorporates the MIMIC-CXR and openi datasets.
5.The logical flow of the text is not clear enough. The introduction highlights the primary flaw of existing VLMs as the hallucination issue, yet the paper does not propose an effective method to eliminate hallucinations. Specifically, the introduction of SAE as a solution lacks persuasiveness. Additionally, the introduction states that 'the generated reports a VLM provides may not be faithful to the underlying computations of the image encoder.' Is there factual evidence to support this claim?

**Questions:**

1.Why not try replicating these indicators using the open-source methods mentioned above?
2.Why not explore two additional open-source datasets?
3.How can the hallucination issue discussed in the article be resolved?

---

> ### Author Response · Authors · 2024-11-18
> **Response 1 Part 1**
>
> We thank the reviewer for their questions and insights. We are glad they found our analyses comprehensive, and our ablations to be thoroughly designed. We are also pleased they found the methodology to be clearly motivated and effective. We have made a number of amendments to our manuscript based on the feedback and respond to the queries below.
>
> > The novelty of the method is not thoroughly demonstrated. While applying Sparse Autoencoders (SAE) in vision-language models (VLMs) represents an innovative effort, the use of SAEs to enhance the interpretability of large language models (LLMs) is already a well-established area [...]
>
> We are training SAEs on vision encoders only rather than VLMs or LLMs.  We believe SAE-Rad to be the first framework to use tools from mechanistic interpretability in order to perform a downstream multi-modal reasoning task. With a significantly smaller dataset and less computational resources for training (a single A6000 NVIDIA GPU), we are already competitive with many state-of-the-art systems based on dense (and much larger) VLMs. Additionally, we have presented a novel sparse autoencoder architecture which achieves superior reconstructions with improved sparsity. OpenAI’s work is contemporaneous to ours - and indeed is currently under review itself at ICLR 2025. Additionally, these works investigate the previous setting of leveraging SAEs for LLM features, rather than investigating interpretable visual features. We have nevertheless cited these works in our related works section.
>
> > Why not try replicating these indicators using the open-source methods mentioned above?
> > The comparison with existing methods is limited. Table 1 only compares three methods: CheXagent, MAIRA-1, and MAIRA-2 [...]
>
> Thank you for this important point. We have **added a number of additional comparator models in Table 7 in Appendix J**. As can be seen, SAE-Rad is strongly competitive with most approaches, outperforming them on several important clinical metrics. We reproduce the table below for convenience:
>
> | Model                               | BL4  | RG-L  | Ma-5  | Ma-14 | Mi-5  | Mi-14 |
> |-------------------------------------|------|-------|-------|-------|-------|-------|
> | GPT-4V [3]                          | 1.9  | 13.2  | 19.6  | 20.4  | 25.8  | 35.5  |
> | LLaVa-Med [3]                       | 1.0  | 13.3  | 16.6  | 15.5  | 22.0  | 27.2  |
> | CvT2Dist [4]                        | 12.7 | 28.6  | -     | 30.7  | -     | 44.2  |
> | LLaVa [4]                           | 1.3  | 13.8  | 17.5  | 15.4  | 23.4  | 22.9  |
> | GPT-4o finetune [4]                 | 17.8 | 32.1  | 43.8  | 33.0  | 52.7  | 48.9  |
> | GPT-4o mini finetune [4]            | 16.2 | **32.2** | 42.0  | 30.8  | 51.8  | 47.6  |
> | R2GenGPT [5]                        | 13.4 | 16.0  | -     | -     | -     | 38.9  |
> | METransformer [5]                   | 12.4 | 29.1  | -     | -     | -     | 31.1  |
> | R2Gen [5]                           | 10.3 | 27.7  | -     | -     | -     | 27.6  |
> | R2GenCMN [5]                        | 17.0 | 19.1  | -     | -     | -     | 27.8  |
> | MSAT [6]                            | 11.1 | -     | -     | -     | -     | 33.9  |
> | KIUT [6]                            | 11.3 | 28.5  | -     | -     | -     | 32.1  |
> | RGRG [7]                            | 12.6 | 26.4  | -     | -     | 54.7  | 44.7  |
> | Flamingo-CXR [8]                    | **29.7** | 10.1  | -     | -     | **58.0** | 51.9  |
> | **SAE-Rad (x64)**                   | 1.9  | 17.1  | **47.2** | **34.3** | 54.4  | **53.2** |

---

> ### Author Response · Authors · 2024-11-18
> **Response 1 Part 2**
>
> > Although the paper notes that NLG Metrics are not sufficiently comprehensive for evaluating radiological reports, the vast majority of report generation studies still rely on NLG Metrics as the primary basis for evaluation.
>
> The current state-of-the-art systems for radiology report generation no longer rely on natural language generation (NLG ) scores as their primary assessment metric for generated reports [1,2,3]. Unfortunately, traditional NLG metrics are insufficient for radiology report generation evaluation as they treat all words equally without accounting for clinical significance [1]. For instance, “there is no pneumonia present” and “pneumonia is not detected” has a 4-gram overlap of 0 but both statements mean the same thing with identical clinical significance. Leveraging radiology-specific clinical metrics based on appropriately pre-specified findings classes [4] and/or calibrated radiology-specific models or error classes [5] are more appropriate. In addition to competitive performance in radiology-specific metrics, we also have a reader study with a specialist radiologist who indicated that our generated reports were of higher quality than other leading open-source state-of-the-art approaches despite lower NLG metrics. In Appendix C, we provide various examples of generated reports which have much different word-order/style to the ground truth (and thus low NLG performance) but that are very high quality reports.
>
> > The quantitative comparisons are conducted using only one public dataset, MIMIC-CXR [...] The paper R2GenGPT, XrayGPT incorporates the MIMIC-CXR and openi datasets.
>
> We leverage MIMIC-CXR as most other models report official test-set performance on this dataset which allows us to compare models more easily [1,6,7,8]. We believe the dataset to be appropriate for our feasibility analysis to demonstrate the utility of the method (given the time and computational constraints).
>
> > The logical flow of the text is not clear enough. The introduction highlights the primary flaw of existing VLMs as the hallucination issue [...]
>
> Thank you for raising this important point. Recent studies have shown that LLMs inherently hallucinate [9], even when well-calibrated [10]. Current VLMs rely on LLMs to generate radiology reports by mapping image latents into the LLM token space. This means that one could never guarantee a hallucination-free system, and consequently there will always be cases where VLMs are not faithful to the underlying computations of an image encoder.
>
> The quality of generated reports is fundamentally limited by the semantic information in the image latents [11]. We therefore propose using an SAE to reverse-engineer image encoder latents into interpretable sparse features. Our approach mitigates hallucinations in three ways: 1) By enabling direct inspection of visual features for each SAE dimension, ensuring faithful descriptions of image latents; 2) By generating reports directly from feature descriptions; 3) By allowing for error correction in feature labelling with human-in-the-loop methods, unlike traditional VLMs.
>
> Additionally, we have developed interactive tooling to identify and eliminate potentially incorrectly labelled features through a user interface.  This tooling is accessible [here](https://scatter-plot-app.vercel.app/).
>
> Thank you for raising this important point of discussion which we have now added to the main manuscript.

---

> ### Author Response · Authors · 2024-11-18
> **Response 1 Part 3**
>
> References:
>
> 1. Bannur, Shruthi, et al. "Maira-2: Grounded radiology report generation." arXiv preprint arXiv:2406.04449 (2024).
> 2. Jean-Benoit Delbrouck, Pierre Chambon, Christian Bluethgen, Emily Tsai, Omar Almusa, and Curtis P Langlotz.Improving the factual correctness of radiology report generation with semantic rewards.*arXiv preprint arXiv:2210.12186*, 2022.
> 3. Feiyang Yu, Mark Endo, Rayan Krishnan, Ian Pan, Andy Tsai, Eduardo Pontes Reis, Eduardo Kaiser Ururahy Nunes Fonseca, Henrique Min Ho Lee, Zahra Shakeri Hossein Abad, Andrew Y Ng, et al.Evaluating progress in automatic chest x-ray radiology report generation.*Patterns*, 4(9), 2023.
> 4. Akshay Smit, Saahil Jain, Pranav Rajpurkar, Anuj Pareek, Andrew Y Ng, and Matthew P Lungren.Chexbert: combining automatic labelers and expert annotations for accurate radiology report labeling using bert.*arXiv preprint arXiv:2004.09167*, 2020.
> 5. Feiyang Yu, Mark Endo, Rayan Krishnan, Ian Pan, Andy Tsai, Eduardo Pontes Reis, Eduardo Kaiser Ururahy Nunes Fonseca, Henrique Min Ho Lee, Zahra Shakeri Hossein Abad, Andrew Y Ng, et al.Evaluating progress in automatic chest x-ray radiology report generation.*Patterns*, 4(9), 2023.
> 6. Tanno, Ryutaro, et al. "Collaboration between clinicians and vision–language models in radiology report generation." *Nature Medicine* (2024): 1-10.
> 7. Wang, Zhanyu, et al. "Metransformer: Radiology report generation by transformer with multiple learnable expert tokens." *Proceedings of the IEEE/CVF Conference on Computer Vision and Pattern Recognition*. 2023.
> 8. Tanida, Tim, et al. "Interactive and explainable region-guided radiology report generation." *Proceedings of the IEEE/CVF Conference on Computer Vision and Pattern Recognition*. 2023.
> 9. Xu, Ziwei, Sanjay Jain, and Mohan Kankanhalli. "Hallucination is inevitable: An innate limitation of large language models." arXiv preprint arXiv:2401.11817 (2024).
> 10. Kalai, Adam Tauman, and Santosh S. Vempala. "Calibrated language models must hallucinate." Proceedings of the 56th Annual ACM Symposium on Theory of Computing. 2024.
> 11. "Bridging the Data Processing Inequality and Function-Space Variational Inference." *ICLR Blogposts 2024*, International Conference on Learning Representations, 2024, https://iclr-blogposts.github.io/2024/blog/dpi-fsvi/. Accessed 18 Nov. 2024.

---

> ### Author Response · Authors · 2024-11-25
>
> Dear reviewer,
>
> Seeing as the rebuttal period is nearly over, we were wondering whether you could assess our rebuttal -- we would love to know what you think!
>
> Thank you for your time,
>
> The authors.

---

> > ### Comment · Reviewer_hUc2 · 2024-11-27
> >
> > Thank you very much for the detailed supplement to the experimental section. In my opinion, retaining NLG-related metrics for radiology report generation remains beneficial, as medical reports need to follow specific medical templates rather than being written in a completely free-form text style. Additionally, the author mentioned that SAE could induce hallucinations in VLMs. Would it be possible to provide related experiments to support this claim?

---

> ### Author Response · Authors · 2024-11-27
>
> We deeply appreciate your response! We completely agree that NLG can be important in some settings, but would argue that in this particular case, findings sections for chest radiographs are produced as free-text paragraphs- we argue that what you really care about is whether or not each statement produced by the system is *clinically accurate*, irrespective of how it's worded (I.e. "no pneumonia" vs "pneumonia not present") should be considered equal in a free-text findings section for a radiograph.
>
> With regards to SAE vs VLM hallucinations, we assume the reviewer in fact means *reduce* hallucinations rather than *induce*? If this is the case, **we have now extended our analysis in Appendix E (under E.1; results)**.  We sub-sampled all sentences from our reader study where the radiologist had made at least one edit, and then manually inspected the radiologist-edited sentences and compared these against the statements provided by the models. We also read the radiologist feedback for their edit categorization. We defined a "hallucination" as the presence of a clinical fact which is either: 1) Not verifiable and/or; 2) Incorrect based on the radiologist feedback.
>
> We summarize the results for the hallucination analysis:
>
> - Overall, we find that 124 (53.4%) of edits were classified as hallucinations, whilst 108 (46.5%) of edits were not classified as hallucinations. Of these edits, SAE-Rad produced the fewest hallucinations, with the baseline generating the most.
> - SAE-Rad produced 50 (64.1%) non-hallucinatory sentences, and 28 (35.9%) sentences that were classified as hallucinations.
> - CheXagent produced 30 (41.1%) non-hallucinatory sentences, and 43 (58.9%) sentences that were classified as hallucinations.
> - The baseline method produced 28 (34.5%) non-hallucinatory sentences, and  53 (65.4%) sentences that were classified as hallucinations.
>
> In summary, SAE-Rad demonstrably produces fewer hallucinations, by a margin of 23.1% when compared to a leading open source CXR foundation model. It produces fewer hallucinations than not when assessed by a specialist radiologist.
>
> Thank you once more for your consideration,
>
> The authors

---

> > ### Comment · Reviewer_hUc2 · 2024-11-29
> >
> > Thank you for your detailed response and the effort you have put into addressing the feedback. While the proposed algorithm demonstrates some interesting ideas, its level of innovation appears limited, and the experimental results do not provide sufficient evidence to strongly support its effectiveness. Based on these considerations, I am inclined to maintain my current score.

---

### Official Review · Reviewer_xYKe · 2024-10-28

**Soundness:** 2
**Presentation:** 3
**Contribution:** 2
**Rating:** 5
**Confidence:** 5

**Summary:**

This paper concentrates on the interpretability of VLMs for radiology report generation. It uses SAE to decompose visual features of an input image and exploits an LLM to distill the entire report into descriptions of SAE features. Finally, by compiling these descriptions, a full report can be obtained.

**Strengths:**

1.This work discusses the interpretability of VLMs in radiology report generation tasks that is a hot, interesting, and critical topic.

2.The authors provide an extensive analysis in the paper and appendix.

**Weaknesses:**

1.As far as storytelling is concerned, the motivation for using SAE for model interpretability is not clear to me in the abstract and introduction. I can find some clues in Section 3, but in general it is not obvious, especially to readers unfamiliar with the topic.

2.Overall, the readability of the paper needs further improvement.

3.In terms of methodology, SAE has been well studied in interpreting LLMs (see https://arxiv.org/pdf/2406.04093; https://openreview.net/pdf?id=XkMrWOJhNd). I understand that the focus of this work is on VLMs, and while I like the topic discussed in this paper, I have to say that it does not seem very novel to me. Also, I note that gated SAE from Rajamanoharan et al., 2024.

4.The following paper utilizes the idea of sparse learning (not limited to SAE) to interpret VLMs: https://arxiv.org/abs/2402.10376. The authors do not mention this work.

5.Regarding experiments, in my opinion, the current results are not very convincing, mainly because of the choice of competing methods. I understand that VLM-based methods should be main competitors. But the authors should also compare their model with other non-VLM approaches, because there are already many. I believe that the study of model interpretability should be based on the fact that the model can work well. In addition, there are more VLM-based, open-source methods for radiology report generation, e.g., RaDialog and R2GenGPT, which the authors have not considered. Note that many VLM and non-VLM approaches are open-source, and the authors can reimplement them and compute all the same metrics for them.

**Questions:**

My main concerns are the novelty of the paper (see Q3 and Q4) and the persuasiveness of experimental results (cf. Q5). In addition, it would be better if the authors can improve the readability of their paper, considering the readers unfamiliar with this topic. I am willing to raise my rating if the authors can address my concerns.

---

> ### Author Response · Authors · 2024-11-18
> **Response 1**
>
> We thank the reviewer for their questions and insights. We are glad they found the our research topic interesting and important, and that they found our analysis to be extensive in both the main manuscript and appendices. We have made a number of amendments to our manuscript based on the feedback and respond to the queries below.
>
> > As far as storytelling is concerned, the motivation for using SAE for model interpretability is not clear to me in the abstract and introduction [...]
>
> Thank you for this comment. We have **amended the abstract and introduction to better motivate the use of SAEs for model interpretability**. In Section 3 of our paper, we introduced SAEs as a method to reverse superposition in order to produce monosemantic disentangled representations. As such, the motivation for using SAEs is explicitly to interpret previously black box representations of models. We have cited a number of related papers that have demonstrated and introduced the use of SAEs for interpretability. The novelty of our work lies in applying SAEs to vision-only encoders (and not a pre-trained vision-language model) and inventing a new method of multi-modal reasoning that we apply to radiology.
>
> > Overall, the readability of the paper needs further improvement.
>
> We have made a number of amendments which we believe simplify the language in a few areas and/or further improve readability. If the reviewer has any additional suggestions, please do let us know.
>
> > In terms of methodology, SAE has been well studied in interpreting LLMs [...]
>
> Thank you for pointing us to these contemporaneous works (indeed, we widely cite the gated-SAE work by Rajamanoharan in Section 3). We would like to highlight that our work is the first to apply SAEs to a vision-only encoder rather than a VLM (or an LLM) in order to perform a multi-modal reasoning task (in this case automated radiology reporting). To the best of our knowledge, our work is the first to use SAE features for any downstream reasoning task.
>
> > The following paper utilizes the idea of sparse learning (not limited to SAE) to interpret VLMs: https://arxiv.org/abs/2402.10376. The authors do not mention this work.
>
> Thank you for pointing us to this paper - **we have added it as a related work**. We would like however to highlight some important differences between our work and this paper. The paper is not used to interpret VLMs. Instead, the method is limited to interpreting CLIP models. The sparse vision representations used in the paper are formed using pre-specified supervised text embeddings from the text encoder. In contrast, our sparse SAE representations are unsupervised and therefore more general, and thus more faithfully represent the underlying model representations. In addition to this, their work requires a LASSO regression at inference time for *each new input*, meanwhile our SAEs are pretrained and do not require this step.
>
> > Regarding experiments, in my opinion, the current results are not very convincing, mainly because of the choice of competing methods [...]
>
> Thank you for your suggestion. We have now **added a new section to compare SAE-Rad to a number of additional techniques in Appendix J**. As can be seen, SAE-Rad is strongly competitive with most approaches, outperforming them on several important clinical metrics. We reproduce the table below for convenience:
>
> | Model                               | BL4  | RG-L  | Ma-5  | Ma-14 | Mi-5  | Mi-14 |
> |-------------------------------------|------|-------|-------|-------|-------|-------|
> | GPT-4V [3]                          | 1.9  | 13.2  | 19.6  | 20.4  | 25.8  | 35.5  |
> | LLaVa-Med [3]                       | 1.0  | 13.3  | 16.6  | 15.5  | 22.0  | 27.2  |
> | CvT2Dist [4]                        | 12.7 | 28.6  | -     | 30.7  | -     | 44.2  |
> | LLaVa [4]                           | 1.3  | 13.8  | 17.5  | 15.4  | 23.4  | 22.9  |
> | GPT-4o finetune [4]                 | 17.8 | 32.1  | 43.8  | 33.0  | 52.7  | 48.9  |
> | GPT-4o mini finetune [4]            | 16.2 | **32.2** | 42.0  | 30.8  | 51.8  | 47.6  |
> | R2GenGPT [5]                        | 13.4 | 16.0  | -     | -     | -     | 38.9  |
> | METransformer [5]                   | 12.4 | 29.1  | -     | -     | -     | 31.1  |
> | R2Gen [5]                           | 10.3 | 27.7  | -     | -     | -     | 27.6  |
> | R2GenCMN [5]                        | 17.0 | 19.1  | -     | -     | -     | 27.8  |
> | MSAT [6]                            | 11.1 | -     | -     | -     | -     | 33.9  |
> | KIUT [6]                            | 11.3 | 28.5  | -     | -     | -     | 32.1  |
> | RGRG [7]                            | 12.6 | 26.4  | -     | -     | 54.7  | 44.7  |
> | Flamingo-CXR [8]                    | **29.7** | 10.1  | -     | -     | **58.0** | 51.9  |
> | **SAE-Rad (x64)**                   | 1.9  | 17.1  | **47.2** | **34.3** | 54.4  | **53.2** |

---

> > ### Comment · Reviewer_xYKe · 2024-11-26
> >
> > I commend the authors for their efforts in addressing my previous comments. However, I maintain reservations regarding the methodological novelty and the significance of the reported improvements.

---

> ### Author Response · Authors · 2024-11-25
>
> Dear reviewer,
>
> Seeing as the rebuttal period is nearly over, we were wondering whether you could assess our rebuttal -- we would love to know what you think!
>
> Thank you for your time,
>
> The authors.

---

### Official Review · Reviewer_ZCEF · 2024-10-30

**Soundness:** 3
**Presentation:** 3
**Contribution:** 3
**Rating:** 8
**Confidence:** 3

**Summary:**

•This paper introduces SAE-Rad, a methodology that uses SAEs to decompose latent vision transformer representations into features, some of which may be human-interpretable. These features are then used to adapt a pretrained LLM to directly generate radiology reports. Ablation studies are performed to evaluate the sparsity elements of the SAE and some qualitative evaluations of the report quality and counterfactual images is also performed.

**Strengths:**

•	The paper has a novel implementation of sparse autoencoders for the task of general text generation (radiology reports are the use case in this study).
•	The quality of the writing is high and the paper is quite easy to comprehend.
•	Multiple examples are provided which helps building a better understanding of the model performance

**Weaknesses:**

•	It is a bit unclear whether the overall goal of the paper is to describe whether SAEs can be used to generate text or to develop a high-quality report generation model. If it is the former, then additional comparisons with more studies image captioning datasets would be better. If it is the latter, then the qualitative reader study would have to be substantially improved, more domain-specific comparisons would be needed, and there could be more domain-specific metrics included. Currently, the paper straddles the area between both fields.
•	For the claim of modularity of image encoders and LLMs, quantitative comparisons would be needed to solidify this argument.
•	Metrics choice: Evaluating report quality is challenging, as is well pointed out. Besides RadFact, there are a few new approaches that are being used. The Rexval and Green scores could be particularly useful here.
•	Qualitative investigation shows that randomly selected feature modes depict to human perceivable concepts. This study would be stronger if a more rigorous analysis of these features were to be performed to better understand what the signifiance of these features is.
•	The ablation study uses the radfact score, which has not been extensively substantiated by the community. More common scores here may be better as comparisons since many other models are not benchmarked with this metric.
•	The entire counterfactual section seems very abrupt and well not justified or explained. What is unsupervised segmentation? This section needs substantially more detail or needs to be removed.
•	The reader study seems very superficial, unfortunately. 30 reports are too small to have concluding thoughts from. Were sentences evaluated or the whole results? What constitutes a significant error? Upon reading the appendix, it seems that only 10 reports are analyzed. It is impossible to draw any conclusions from this. Evaluating sentence by sentence also seems worse than evaluation of the whole report.

**Questions:**

•	Eq: 4 Is there an empirical technique for understanding what an optimal number for /pi_gate should be compared to /h_hate?
•	Difference between SAE and SAE-Rad #2: What would the impact of enforcing decoder sparsity be on the encoder itself? One could likely posit that it could further encourage encoder sparsity?
•	Difference between SAE and SAE-Rad #4: The choice of the L0 norm is seemingly heuristic? Could this be relaxed to have more firing features to better represent entangled concepts? I also wonder if there is something to be gained from exploring the relationships between this optimization paradigm and beta-vaes that expicitly force some level of disentanglement of feature sets.
•	How is set R(i) chosen when creating natural language representations of firing features? I suspect this would substantially depend on the size of R? Moreover, i wonder how representations learned from your SAE vary compared to the usual VAE-style methods. Are SAE grouped images also grouped with traditional VAEs?
•	Figure 2 is very interesting. Should we be able to hypothesize whether the learned firing features should be coarse grained (like devices/tubes) or fine grained (specific sided pleural effusion)?
•	In Figure 3, how is it that SAE-Rad does not include any comparisons with an image prior (the last two sentences in the ground truth report)?
•	Ablation study: are there specific differences in compute/performance with higher expansion factors? Moreover, what is unique about x64 that makes it better than 128 or 32?
•	What was the best performing model that was used? From the ablation it seems like the 64 features? If so, title of the paper should be edited.
•	What is the performance of MAIRA models with the same amount of data as SAERad?

---

> ### Author Response · Authors · 2024-11-18
> **Response 1 Part 1**
>
> We thank the reviewer for their questions and insights. We are glad they found the framework to be novel, the quality of our writing to be high, and feel we provided an appropriate number of examples to understand our results. We have made a number of amendments to our manuscript based on the feedback (including a novel proof based on one of the discussion points around sparsity), and respond to the remaining queries below.
>
> > It is a bit unclear whether the overall goal of the paper is to describe whether SAEs can be used to generate text or to develop a high-quality report generation model [...]
>
> This work is a feasibility study exploring whether mechanistic interpretability can support automated pipelines to generate image descriptions. We chose radiology report generation as a challenging, clinically relevant case study. We selected metrics based on the assessments made for other state-of-the-art radiology systems - thus we reported both general lexical metrics and domain-specific radiology metrics to allow for comparison with other frameworks in the literature (e.g., [1,2]). We appreciate the reviewer’s point on the size of the reader study, however would highlight that this is a highly detailed analysis which was designed based on the reader assessment carried out on reports generated by the current state-of-the-art system [1]. Nevertheless, working with our specialist radiologist, we aim to double the size of the reader study by the end of the rebuttal period, and will include the expanded analysis in the final manuscript. This will mean the size of the reader study will be three times greater than the state-of-the-art study which had comparable experimental design. Finally, we have added a number of **additional domain-specific comparisons with other approaches in Appendix J.**
>
> > For the claim of modularity of image encoders and LLMs, quantitative comparisons would be needed to solidify this argument.
>
> Thank you for this point — we simply meant that different language models could be used to label the SAE features and/or synthesise the activated features for a given image into a radiology report. This would simply be a user-adjustable design choice.
>
> > Metrics choice: Evaluating report quality is challenging, as is well pointed out. Besides RadFact, there are a few new approaches that are being used [...]
>
> Thank you for highlighting other LLM-based assessments, namely the Chexprompt and Green scores. While these metrics represent promising approaches, we selected RadFact as our primary LLM-as-a-judge metric to align with the current state-of-the-art system, MAIRA-2 [1], which also uses RadFact. Additionally, RadFact offers a robust framework for evaluating clinical accuracy and completeness, which we deemed most relevant for our study. Computational cost and practical constraints also influenced our decision. We aim to investigate other promising assessment approaches in future work.
>
> > Qualitative investigation shows that randomly selected feature modes depict to human perceivable concepts. This study would be stronger if a more rigorous analysis of these features were to be performed to better understand what the signifiance of these features is.
>
> Thank you for this comment - we are excited to share an **interactive UI which visualises a UMAP embedding of our SAE features [here](https://scatter-plot-app.vercel.app/)**. This allows you to explore the significance of the features and also to investigate their geometry [3]. We find a number of highly interpretable feature clusters including breathing tube instrumentation ([anonymous img 1](https://imgur.com/BPD7R0s)), the absence of lung field pathology ([anonymous img 2](https://imgur.com/Feio2uJ)), the presence of pacemakers ([anonymous img 3](https://imgur.com/yOsY6HA)) and post-surgical contexts ([anonymous img 4](https://imgur.com/OnP5hoU)). We additionally note interesting feature clusters which relate to abstract concepts such as image orientation ([anonymous img 5](https://imgur.com/mLvOZds)). We look forward to further exploring these patterns in future work. We discuss this in further detail in **Appendix I**.

---

> ### Author Response · Authors · 2024-11-18
> **Response 1 Part 2**
>
> > The ablation study uses the radfact score, which has not been extensively substantiated by the community. More common scores here may be better as comparisons since many other models are not benchmarked with this metric.
>
> This is a nice suggestion. We have added additional clinically relevant evaluations for the ablation study in **Appendix H**, which we reproduce in summary form below for convenience. In summary, these additional experiments support our previous findings.
>
> | Model             | RadFact F1 | BL4  | RG-L  | MTR  | RGER | Ma-5  | Ma-14 | Mi-5  | Mi-14 |
> |--------------------|------------|------|-------|-------|------|-------|-------|-------|-------|
> | ×32 dense         | 30.22      | 1.5  | *18.6*| 23.3  | *19.5* | 51.8  | 33.2  | 55.0  | 52.6  |
> | ×64 dense         | 29.74      | *1.9*| **19.5** | *23.9*| *19.5* | **57.8** | 31.9  | *58.7*| *55.5* |
> | ×128 dense        | 29.56      | 1.7  | 16.1  | 22.8  | 17.3  | 46.9  | **33.9** | 57.5  | 53.7  |
> | ×32               | 29.46      | 2.2  | 16.2  | 23.7  | 16.6  | 44.2  | 25.3  | 53.5  | 48.8  |
> | ×64               | **33.83**  | **2.4** | 17.1  | **24.4** | **20.7** | *54.7* | *33.6* | **58.9** | **56.8** |
> | ×128              | *32.18*    | 1.8  | 16.4  | 24.1  | 18.9  | 45.2  | 25.7  | 55.8  | 49.8  |
>
> > The entire counterfactual section seems very abrupt and well not justified or explained. What is unsupervised segmentation? This section needs substantially more detail or needs to be removed.
>
> Thank you for this comment. The counterfactual section aims to assess whether the features learned by our sparse autoencoder correspond meaningfully to their spatial locations in x-rays. To achieve this, we trained a vision decoder that maps interventions in SAE-space back to the image space, allowing us to evaluate the alignment of latent features with x-ray regions. We agree that additional detail is necessary and have **expanded the motivations and experimental procedures in Appendix D**. We have also revised the main manuscript to provide clearer justification for the section and renamed it to better reflect its focus.
>
> > The reader study seems very superficial [...]
>
> We have asked our expert radiologist to double the current size of the reader study as mentioned above. **We have now updated our reader study to include these results in Appendix E**. We note our study size now matches the size of previously reported studies with the same study design [1: Appendix Section F.2], and triples the number of findings assessed. SAE-Rad continues to have the highest quality reports of the approaches assessed. Indeed, we based our reader study methodology on the assessment performed in the MAIRA-2 system paper. In this type of assessment, each individual sentence (or ‘finding’) is evaluated individually by a trained specialist radiologist. This allows us to assess each finding along multiple dimensions (e.g., is this finding too detailed/not detailed enough? Does the finding reference appropriate anatomy? Is the anatomy localised correctly/should it refer to the contralateral side?). With regards to significant vs non-significant errors, the specialist can distinguish between major clinical errors such as missing a tension pneumothorax (fatal) and errors which have minor clinical implications (e.g., misclassifying a mild pneumonia as moderate - as the treatment is often the same in both cases). We believe evaluations at the phrase level enable higher-level aggregates and quantifications, provide a record of concrete corrections, and deepen understanding of what constitutes “good quality” in reporting outputs.
>
> ## Questions
>
> > Eq: 4 Is there an empirical technique for understanding what an optimal number for /pi_gate should be compared to /h_hate?
>
> Pi_gate is a learnable pre-activation of h_gate, which is an element-wise heavistep function that attempts to learn which features are activated by a given input. Pi_gate is learnt by stochastic mini-batch gradient descent, and the step-function is fixed.
>
> > Difference between SAE and SAE-Rad #2: What would the impact of enforcing decoder sparsity be on the encoder itself? One could likely posit that it could further encourage encoder sparsity?
>
> Thank you for this insightful question. Enforcing sparsity in the decoder (e.g., by setting column weights to 0) would effectively remove that feature from the SAE, akin to using a smaller expansion factor. Our current design ensures encoder sparsity via the L1 regularization term in the loss. The L2 norm of the decoder weights is included in the sparsity loss simply to prevent the SAE from exploiting weight scaling between the encoder and decoder during training, which is described in other works [4,5]. However, we note this loss is equivalent to the ‘vanilla’ SAE loss from [6]. **We have derived a novel proof of this, which we add to Appendix K**.

---

> ### Author Response · Authors · 2024-11-18
> **Response 1 Part 3**
>
> > How is set R(i) chosen when creating natural language representations of firing features? [...]
>
> For a given SAE feature, we select the 10 highest-activating images from our training data (those with the largest activation magnitudes in that SAE dimension). We then extract the associated radiology reports written by human experts. The choice of 10 reports is a heuristic, but this number can be adjusted as a tunable hyperparameter.
>
> SAE latents form a sparse dictionary, meaning each input image activates only a small subset of SAE features. This design aligns with the superposition and linear representation hypotheses discussed in Section 3.1, leading us to expect SAE latents to be more monosemantic (i.e., tied to a single interpretable concept) than the features learned in a VAE-style approach. However, further experiments would be needed to evaluate whether image groupings in SAEs align with those from traditional VAEs, and this represents an interesting line of future work.
>
> >  Difference between SAE and SAE-Rad #4: The choice of the L0 norm is seemingly heuristic? Could this be relaxed to have more firing features to better represent entangled concepts? [...]
>
> Thank you for this interesting question. The L0 norm is indirectly controlled via the L1 regularisation term in the loss. SAE training optimises the trade-off between sparsity (L0 norm) and reconstruction loss, which lies on a pareto frontier: reducing sparsity (higher L0) can lower reconstruction loss but may degrade task performance.
>
> As shown in our ablation study (Section 5.2), training less sparse (higher L0) SAEs degrades performance for our task, while overly sparse (lower L0) configurations also perform worse. This suggests an optimal level of sparsity for our task.
>
> Unlike SAEs, β-VAEs do not explicitly enforce sparsity in latent representations and thus do not leverage the superposition and linear representation hypotheses. However, investigating potential connections between SAE and β-VAE latents (and how disentanglement paradigms interact with sparsity) is an interesting direction for future work.
>
> > Figure 2 is very interesting. Should we be able to hypothesize whether the learned firing features should be coarse grained (like devices/tubes) or fine grained (specific sided pleural effusion)?
>
> Thank you! Empirically, we find that SAEs are capable of learning both coarse and fine-grained features. The level of granularity is determined by the expansion factor. Contemporaneous work has shown that feature absorption can occur as the expansion factor is changed [7, 8]. We further discuss the effect of different expansion factors in our ablation study (**Section 5.2**).
>
> > In Figure 3, how is it that SAE-Rad does not include any comparisons with an image prior (the last two sentences in the ground truth report)?
>
> The ground truth is generated by an expert human radiologist who has access to *all* prior X-rays taken for a given patient. This allows them to make temporal statements such as ‘the pacemaker is still present’. In our pipeline — we do not currently use prior images directly to generate a radiology report. We simply generate a report of the current image, which means that we are less likely to capture temporal imaging information. This is a feature we are hoping to develop in future work.
>
> > Ablation study: are there specific differences in compute/performance with higher expansion factors? Moreover, what is unique about x64 that makes it better than 128 or 32?
>
> The compute scales linearly with expansion factor. In **Appendix H** , we analyse the performance differences across three different expansion factors based on the MSE and L0. However, we chose the 64 expansion factor for the main analysis based on the fact that it had the highest RadFact score. We have **additionally expanded Appendix H** to include new clinical metrics supporting these findings.
>
> > What was the best performing model that was used? From the ablation it seems like the 64 features? If so, title of the paper should be edited.
>
> We would like to clarify that the number in our title ‘15 features’ represents the L0 norm of our best performing models. That is, on average, each X-ray activates approximately 15 SAE features, which is enough to produce very high quality radiology reports. The number 64 refers instead to the expansion factor that we multiply the input dimension by.
>
> > What is the performance of MAIRA models with the same amount of data as SAERad?
>
> We are unfortunately unable to conduct this analysis as the training scripts for MAIRA-2 are not released — additionally, this would be very compute intensive as it is a substantially larger VLM, wheres our SAEs can be trained on a single GPU. We discuss these differences further in **Appendix F**.

---

> ### Author Response · Authors · 2024-11-18
> **Response 1 Part 4**
>
> References:
>
> 1. Bannur, Shruthi, et al. "Maira-2: Grounded radiology report generation." arXiv preprint arXiv:2406.04449 (2024).
> 2. Chen, Zhihong, et al. "Chexagent: Towards a foundation model for chest x-ray interpretation." arXiv preprint arXiv:2401.12208 (2024).
> 3. Mendel, Jake. "SAE Feature Geometry Is Outside the Superposition Hypothesis." *AI Alignment Forum*, 24 June 2024, [www.alignmentforum.org/posts/MFBTjb2qf3ziWmzz6/sae-feature-geometry-is-outside-the-superposition-hypothesis](http://www.alignmentforum.org/posts/MFBTjb2qf3ziWmzz6/sae-feature-geometry-is-outside-the-superposition-hypothesis).
> 4. "Scaling Monosemanticity." *Transformer Circuits*, 2024, https://transformer-circuits.pub/2024/scaling-monosemanticity/. Accessed 17 Nov. 2024.
> 5. Rajamanoharan, Senthooran, et al. "Improving dictionary learning with gated sparse autoencoders." arXiv preprint arXiv:2404.16014 (2024).
> 6. Cunningham, Hoagy, et al. "Sparse autoencoders find highly interpretable features in language models." arXiv preprint arXiv:2309.08600 (2023).
> 7. Chanind, Hrdkbhatnagar, TomasD, and Joseph Bloom. "Toy Models of Feature Absorption in SAEs." *LessWrong*, 7 Oct. 2024, [www.lesswrong.com/posts/kcg58WhRxFA9hv9vN/toy-models-of-feature-absorption-in-saes](http://www.lesswrong.com/posts/kcg58WhRxFA9hv9vN/toy-models-of-feature-absorption-in-saes). Accessed 17 Nov. 2024.
> 8. Chanin, David, et al. "A is for Absorption: Studying Feature Splitting and Absorption in Sparse Autoencoders." arXiv preprint arXiv:2409.14507 (2024).

---

> ### Author Response · Authors · 2024-11-25
>
> Dear reviewer,
>
> Seeing as the rebuttal period is nearly over, we were wondering whether you could assess our rebuttal -- we would love to know what you think!
>
> Thank you for your time,
>
> The authors.

---

> > ### Comment · Reviewer_ZCEF · 2024-11-26
> > **Response to official comment**
> >
> > Thank you to the authors for providing the very substantial response to both the weaknesses and the questions that were raised during the review. I am sufficiently happy with the quality and the rigor of the responses that have been provided. I only have two minor comments that can hopefully be addressed in the case of the eventual camera-ready paper or for a future submission.
> >
> > > Metrics:
> >
> > It would still be beneficial to add some of the other up-and-coming metrics that are used for radiology report evaluation. Showing consistency of results across metrics can considerably strengthen the claims made in the manuscript.
> >
> > > The interactive viewer:
> >
> > This is definitely a great tool. The author would encourage the authors to incorporate this in the corresponding GitHub repo or as a standalone tool alongside the paper.
> >
> > Based on the quality of the response, I have raised the score from a 6 to an 8.

---

### Official Review · Reviewer_EGTD · 2024-11-03

**Soundness:** 3
**Presentation:** 3
**Contribution:** 3
**Rating:** 5
**Confidence:** 5

**Summary:**

This work combined a SAE vision encoder, an LLM and the linked highest activating features, to generate radiology report. The authors used the public MIMIC-CXR dataset for this work.

**Strengths:**

The computational framework seems reasonable. Each component is pretty well established.

**Weaknesses:**

However, the originality of the methods is very marginal. The algorithmic or methodological innovation is low.

**Questions:**

Not sure why the authors did not other additional datasets for independent validation?

---

> ### Author Response · Authors · 2024-11-18
> **Response 1**
>
> > However, the originality of the methods is very marginal. The algorithmic or methodological innovation is low.
>
> We believe SAE-Rad to be the first framework to use tools from mechanistic interpretability in order to perform a downstream multi-modal reasoning task. With significantly less data (e.g., no prior imaging studies, and an overall smaller dataset), and fewer computational resources for training (a single A6000 NVIDIA GPU) we are already competitive with much larger state-of-the-art systems based on dense VLMs. Additionally, we present a novel sparse autoencoder architecture which achieves better input reconstructions with lower sparsity. We are the first to demonstrate the utility of inherently interpretable techniques for reasoning in a medical dataset, and would very much be grateful if you would consider revising your score.
>
> > Not sure why the authors did not other additional datasets for independent validation?
>
> We use the official test split from the MIMIC-CXR dataset for evaluation as this is the most commonly assessed evaluation dataset for the task of radiology report generation and thus  allows us to perform a standardised comparison to other systems in the literature [1].
>
> References:
>
> 1. Bannur, Shruthi, et al. "Maira-2: Grounded radiology report generation." arXiv preprint arXiv:2406.04449 (2024).

---

### Official Review · Reviewer_kgWG · 2024-11-09

**Soundness:** 2
**Presentation:** 3
**Contribution:** 2
**Rating:** 5
**Confidence:** 4

**Summary:**

The paper proposes a novel radiology report generation framework (SAE-Rad) based on Sparse-Autoencoder (SAE). It decomposes latent image features into human-interpretable features, distils these features into radiological descriptions and compiles them into full report. The experiment results on MIMIC-CXR show the leading report generation performance and efficient training computation.

**Strengths:**

## Strengths
- The public dataset, codes, and implementation details are provided for reproducibility.
- Method novelty of some improvements on gated-SAE, which achieves better sparcity and reconstruction quality.
- Enhancing the model interpretability by SAE decomposing image latent features into radiological descriptions.
- Comprehensive evaluation metrices and leading performance in MIMIC-CXR Chest X-ray report generation.

**Weaknesses:**

## Weaknesses and Questions
- The motivation of the method is still not well explained. In 4.1, the authors mention four differences between the gated SAE and SAE-Rad, but I do not find enough explanation about what is the advantage of these differences and why these differences are important for image-to-text generation. Furthermore, I would like to know whether the entire SAE-Rad framework enhances the general image-to-text generation tasks, or it is specifically designed for radiology report generation? If SAE-Rad is mainly designed for report generation, then the authors should also explain the specific advantages of the four differences in SAE-Rad and the sparse feature interpretability method in radiology report generation task.
- No ablation study about the improved SAE loss (Eq. 9). Since the authors make some changes in SAE loss, as described in Eq. 9 and the four differences, there should be some ablation studies showing the effectiveness of this novel SAE loss, compared with other losses used in SAE training.
- There is no discussion for the co-occurence issue in medical report generation. In medical images, multiple diseases or visual markers sometimes have high correlation and frequently shown up together in one image, so this may cause some false findings in the generated reports. Since the proposed SAE-Rad report generation model is based on image latent feature decomposition and activated feature descriptions, so it's important to understand if the feature representations of co-occurence visual markers are fully decomposed and disentangled before generating any feature interpretation descriptions from LLMs. Please provide discussions about the co-occurence issue and the potential limitations of it.
- The number of comparison methods is not enough for a fully demonstration of the effectiveness of the proposed method. In Table 1, only CheXagent is compared in *Current study*, and MAIRA as upper bound. However, since radiology report generation using MIMIC-CXR has been widely explored in recent years, there are much more methods that could be used for comparison. I think the authors should provide more comparison results with some other recent leading methods for a stronger evaluation.
- The authors use RadFact as one of the evaluation metrics since it is robust without relying on pre-specified error types or specialized models. But in Table 1, SAE-Rad only gains 0.3 in RadFact compared to CheXagent, which is a marginal improvement, and also a large gap to the upper bound; in the same time, other clinical metrics show large increases. So how to explain this discrepancy in these metrics? Does it mean RadFact score is not sensitive enough?
- The proposed method is based on the sparse image features (feature directions and associated activations) extracted from SAE-Rad, but the clustering and distributions of the extracted features of all chest diseases/markers are not visualized or discussed. The sparse feature distribution in latent space is important to assess the feature clustering and latent space training, where feature directions of similar visual markers should be clustered closely while different visual markers should push apart their feature directions. I recommand the authors visualize the sparse feature representations of each diseases/markers using T-SNE demo to illustrate the sparse feature clustering.

**Questions:**

The proposed report generation method of the paper has some novelty in SAE and activated feature interpretability, and the result also shows leading generation performance. But some important differences of the method are not fully explained and evaluated, and some common issues in report generation such as co-occurence and latent feature clustering are not discussed, which weaken the insight of the paper. In conclusion, I think the paper is on the boarderline so I currently rate the paper as "borderline accept". I may adjust my rating based on the authors' reponse during discussion period.

---

> ### Author Response · Authors · 2024-11-18
> **Response 1 Part 1**
>
> We thank the reviewer for their questions and insights. We are glad they found our method novel and reproducible, and our evaluations to be comprehensive. We have made a number of amendments to our manuscript based on the feedback, and respond to the queries below.  Based on one of the discussion points, we have also built an interactive web app to explore SAE feature geometry, which we link below.
>
> > The motivation of the method is still not well explained. In 4.1, the authors mention four differences between the gated SAE and SAE-Rad, but I do not find enough explanation about what is the advantage of these differences and why these differences are important [...]
>
> SAEs must learn a sparse dictionary to reconstruct image latents accurately. To achieve this, we optimise the pareto trade-off between reconstruction loss (measured by mean-squared error) and sparsity (measured by the L0 norm). The introduced differences improve the pareto efficiency of SAEs, as validated in **Appendix B**, where we conduct an ablation study on the revised SAE loss (Eq. 9). We have now updated the manuscript to highlight this analysis in the main text.
>
> SAE-Rad is a general multi-modal reasoning framework designed to enhance vision encoder interpretability by breaking down image latents into human-interpretable features. While SAE-Rad can be broadly applied, it requires granular image descriptions (e.g., reference radiology reports) for automated feature labelling. We therefore use the challenging task of automated radiology report generation as a case-study for this approach.
>
> > There is no discussion for the co-occurence issue in medical report generation [...]
>
> Thank you for highlighting this insightful point. We agree that latent representations can encode spuriously correlated information, which SAEs may struggle to disentangle into distinct features. Furthermore, a phenomenon known as "feature absorption" [1] can occur, where information that should be represented as separate concepts is instead subsumed into a single SAE dimension. Whilst feature absorption has recently been noted in SAEs, its aetiology has not yet been well explained. These issues could impact the decomposition of co-occurring visual markers. **We have added a further discussion of these issues into the manuscript.**
>
> > The number of comparison methods is not enough for a fully demonstration of the effectiveness of the proposed method [...]
>
> We have added a new section (**Appendix J**) comparing SAE-Rad to 14 alternative approaches for automated report generation. We reproduce the table below for convenience. As can be seen, SAE-Rad broadly outperforms them across all clinical metrics.
>
> | Model                               | BL4  | RG-L  | Ma-5  | Ma-14 | Mi-5  | Mi-14 |
> |-------------------------------------|------|-------|-------|-------|-------|-------|
> | GPT-4V [3]                          | 1.9  | 13.2  | 19.6  | 20.4  | 25.8  | 35.5  |
> | LLaVa-Med [3]                       | 1.0  | 13.3  | 16.6  | 15.5  | 22.0  | 27.2  |
> | CvT2Dist [4]                        | 12.7 | 28.6  | -     | 30.7  | -     | 44.2  |
> | LLaVa [4]                           | 1.3  | 13.8  | 17.5  | 15.4  | 23.4  | 22.9  |
> | GPT-4o finetune [4]                 | 17.8 | 32.1  | 43.8  | 33.0  | 52.7  | 48.9  |
> | GPT-4o mini finetune [4]            | 16.2 | **32.2** | 42.0  | 30.8  | 51.8  | 47.6  |
> | R2GenGPT [5]                        | 13.4 | 16.0  | -     | -     | -     | 38.9  |
> | METransformer [5]                   | 12.4 | 29.1  | -     | -     | -     | 31.1  |
> | R2Gen [5]                           | 10.3 | 27.7  | -     | -     | -     | 27.6  |
> | R2GenCMN [5]                        | 17.0 | 19.1  | -     | -     | -     | 27.8  |
> | MSAT [6]                            | 11.1 | -     | -     | -     | -     | 33.9  |
> | KIUT [6]                            | 11.3 | 28.5  | -     | -     | -     | 32.1  |
> | RGRG [7]                            | 12.6 | 26.4  | -     | -     | 54.7  | 44.7  |
> | Flamingo-CXR [8]                    | **29.7** | 10.1  | -     | -     | **58.0** | 51.9  |
> | **SAE-Rad (x64)**                   | 1.9  | 17.1  | **47.2** | **34.3** | 54.4  | **53.2** |
>
> > The authors use RadFact as one of the evaluation metrics since it is robust without relying on pre-specified error types or specialized models [...]
>
> Thank you for this point. RadFact has some possible limitations. It relies on an LLM to extract discrete statements from reports, which may introduce inaccuracies. Similarly, bi-directional entailment verification can be challenging for the LLama3 70B model due to complex medical reasoning and medical jargon, as well as known issues like the "reversal curse" [2]. Additionally, RadFact is strict in its comparisons, marking any additional generated information absent from the reference report as incorrect, even if it is clinically accurate. We have added a detailed discussion of these limitations, including illustrative examples, in **Appendix G**.

---

> ### Author Response · Authors · 2024-11-18
> **Response 1 Part 2**
>
> > The proposed method is based on the sparse image features (feature directions and associated activations) extracted from SAE-Rad, but the clustering and distributions of the extracted features of all chest diseases/markers are not visualized or discussed [...]
>
> This is a nice suggestion. Understanding the geometry of SAE features is an active and exciting area of research [3]. Currently, research labs are building interactive tooling to explore this geometry using UMAP dimensionality reduction [4]. Based on your feedback we have **built an interactive UI to visualise and explore the SAE-Rad features using a UMAP [here](https://scatter-plot-app.vercel.app/)**. We find a number of highly interpretable feature clusters including breathing tube instrumentation ([anonymous img 1](https://imgur.com/BPD7R0s)), the absence of lung field pathology ([anonymous img 2](https://imgur.com/Feio2uJ)), the presence of pacemakers ([anonymous img 3](https://imgur.com/yOsY6HA)) and post-surgical contexts ([anonymous img 4](https://imgur.com/OnP5hoU)). We additionally note interesting feature clusters which relate to abstract concepts such as image orientation ([anonymous img 5](https://imgur.com/mLvOZds)). We are looking forward to further exploring these interactive visualisations. We have **added a description of this approach to a new section (Appendix I)**.
>
> References:
>
> 1. Chanin, David, et al. "A is for Absorption: Studying Feature Splitting and Absorption in Sparse Autoencoders." arXiv preprint arXiv:2409.14507 (2024).
> 2. Berglund, Lukas, et al. "The reversal curse: Llms trained on" a is b" fail to learn" b is a"." arXiv preprint arXiv:2309.12288 (2023).
> 3. Mendel, Jake. "SAE Feature Geometry Is Outside the Superposition Hypothesis." *LessWrong*, 24 June 2024, [www.lesswrong.com/posts/MFBTjb2qf3ziWmzz6/sae-feature-geometry-is-outside-the-superposition-hypothesis](http://www.lesswrong.com/posts/MFBTjb2qf3ziWmzz6/sae-feature-geometry-is-outside-the-superposition-hypothesis).
> 4. Neuronpedia. "Open Interpretability Platform." *Neuronpedia*, 2024, [www.neuronpedia.org/](http://www.neuronpedia.org/).

---

> ### Author Response · Authors · 2024-11-25
>
> Dear reviewer,
>
> Seeing as the rebuttal period is nearly over, we were wondering whether you could assess our rebuttal -- we would love to know what you think!
>
> Thank you for your time,
>
> The authors.

---

> ### Comment · Reviewer_kgWG · 2024-11-26
> **comments**
>
> I have read other reviewers comments and authors' feeback which is highly appreciated. Generally speaking, this should be a very practical problem about imaging disgnosis to solve. People stuck with the recent LLM development and the availability of public datasets on chest x-ray with both images and reports. From the reported results, it is very questionable to see if the line of technical development will ever reach a level to help patients. Maybe the whole solution space exists essential problems. There are some nolvety on using SAE and activated feature interpretability but I feel it may not be sufficient. So I may lower my score after serious consideration.

---

> ### Author Response · Authors · 2024-11-26
>
> Thank you for your comment. We would simply stress that our approach either matches - or indeed outperforms - many of the state-of-the-art approaches across important clinical metrics, whilst having two additional distinct advantages:
>
> 1. Being more interpretable by design - Indeed, we believe this work to be the first to use a technique from mechanistic interpretability for any multi-modal reasoning task.
> 2. Being substantially more computationally efficient to train. Our models can be trained on a single GPU as opposed to upwards of 32 A100 cards for some of the models included in our comparisons.
>
> Our approach achieves these results with as little as 23% the size of the datasets used in competing approaches. We therefore strongly believe that an interpretable, accurate, and more compute efficient system such as ours absolutely can reach a level so as to be genuinely, tangibly helpful to patients in a real-world setting. We agree that perhaps there may be essential problems within the whole solution space, but we think this a good approach to take to resolving some of them.

---

### Author Response · Authors · 2024-11-18

We thank all of the reviewers for their time and detailed reviews. We appreciate their recognition of the novelty and reproducibility of our approach (**kgWG** & **ZCEF**), the high quality of our writing (**ZCEF**), and the comprehensiveness of our evaluations across both the main manuscript and appendices (**xYKe** & **kgWG**). We also thank the reviewers for highlighting the importance and relevance of the research topic (**xYKe**), the clear motivation behind our method (**hUc2**) and our thorough experimental design (**hUc2**).

In response to their feedback, we have made several amendments to the manuscript, which we believe has significantly improved it’s quality. For instance:
- Based on a discussion point around SAE feature clustering, we have developed an interactive web app to visualize SAE feature geometry [link](https://scatter-plot-app.vercel.app/).
- We have included a novel proof clarifying the relationship between loss functions across SAE architectures in Appendix K.
- We have extended our ablation study in Appendix H.
- We have added a larger number of methods we compare against in Appendix J.

We hope these amendments and our additional analyses address the reviewers’ feedback and thank them again for their valuable insights and time.

---

### Author Response · Authors · 2024-11-23

Dear reviewers,

We would like to thank you again for your insightful reviews. We would be keen for you to assess our rebuttals, which we are very excited for you to see -- please do let us know what you think!

Thank you for your time and consideration,

The authors.

---

### Meta-Review · Area_Chair_pWVr · 2024-12-21

**Metareview:**

The paper introduces SAE-Rad, a novel framework for radiology report generation that combines Sparse Autoencoders (SAEs) with pre-trained language models to produce interpretable, high-quality reports while maintaining computational efficiency. Reviewers acknowledged the potential of applying mechanistic interpretability techniques to vision encoders and appreciated the robust evaluation and interactive tooling provided. However, concerns were raised regarding the limited methodological novelty, as the approach heavily builds on existing SAE research, and the lack of comprehensive comparisons with non-VLM and additional VLM-based baselines. Despite strong empirical results in some metrics, questions persisted about the broader applicability and the system's improvements over existing methods. While the paper presents valuable contributions, the reservations regarding its technical innovation across multiple reviewers unfortunately still remain.

**Additional Comments On Reviewer Discussion:**

During the rebuttal period, reviewers raised significant concerns about the technical innovation and experimental validity of SAE-Rad, focusing on its limited methodological novelty and the lack of comprehensive comparisons with alternative approaches. The authors responded with expanded comparisons, additional visualizations, and detailed clarifications on issues such as co-occurrence and hallucinations. However, reviewer kgWG lowered their score, citing unresolved concerns about the technical contributions, while ZCEF maintained a high score of 8, arguing for acceptance based on the work's alignment with the technical focus of ICLR rather than immediate clinical applicability. Reviewer EGTD's feedback was not considered due to poor review quality. While I agree that the technical contributions are more critical than clinical impact for this venue, the consistent concerns from reviewers about the innovation and validity of the methodology ultimately led to a rejection decision.

---

### Decision · Program_Chairs · 2025-01-22

Reject